# Does Vector Quantization Fail in Spatio-Temporal Forecasting? Exploring a Differentiable Sparse Soft-Vector Quantization Approach

## Abstract

Spatio-temporal forecasting is crucial in various fields and requires a careful balance between identifying subtle patterns and filtering out noise. Vector quantization (VQ) appears well-suited for this purpose, as it quantizes input vectors into a set of codebook vectors or patterns. Although VQ has shown promise in various computer vision tasks, it surprisingly falls short in enhancing the accuracy of spatio-temporal forecasting. We attribute this to two main issues: inaccurate optimization due to non-differentiability and limited representation power in hard VQ. To tackle these challenges, we introduce Differentiable Sparse Soft-Vector Quantization (SVQ), the first VQ method to enhance spatio-temporal forecasting. SVQ balances detail preservation with noise reduction, offering full differentiability and a solid foundation in sparse regression. Our approach employs a two-layer MLP and an extensive codebook to streamline the sparse regression process, significantly cutting computational costs while simplifying training and improving performance. Empirical studies on five spatio-temporal benchmark datasets show SVQ achieves state-of-the-art results, including a 7.9% improvement on the WeatherBench-S temperature dataset and an average MAE reduction of 9.4% in video prediction benchmarks (Human3.6M, KTH, and KittiCaltech), along with a 17.3% enhancement in image quality (LPIPS). Code is publicly available at https://anonymous.4open.science/r/SVQ-Forecasting.

## 1 Introduction

Spatio-temporal forecasting is pivotal in numerous domains ranging from environmental monitoring to urban planning, where precisely predicting future dynamics is crucial. The journey to refine forecasting methods has spanned from traditional feature engineering to the latest explorations in deep learning. Among various methodologies explored, Vector Quantization (VQ) has distinguished itself primarily in computer vision tasks, showcasing its ability to compress high-dimensional vectors into a compact, discrete form that maintains significant fidelity to the original information. Historically rooted in signal processing, VQ's breakthrough came with its application in image processing advancements like the Vector Quantised-Variational AutoEncoder (VQ-VAE) van den Oord et al. (2017), which set a precedent in generating high-quality images by learning efficient representations of complex distributions.

While VQ has proven effective and has become a nearly default approach in computer vision generation tasks, its potential applications in spatio-temporal forecasting remain less explored. Considering the noise reduction capabilities of VQ, along with the similarities between image/video generation and spatio-temporal forecasting, one could infer that VQ would positively impact the latter. However, our review of existing studies reveals that few have successfully enhanced spatio-temporal forecasting performance using VQ techniques. **Our empirical analysis of recent state-of-the-art VQ methods discovered that all of them fell short of expectations, often degrading the performance of baseline forecasting models rather than providing the anticipated improvements**, as illustrated in Figure 1 and detailed in Table 4. They consistently exert a significant negative influence on the final MSE or MAE regression accuracy.

We believe that the similarity between image/video generation and spatio-temporal forecasting underscores the potential of VQ, but the problem lies in the inherent dynamic nature of spatio-temporal data. The complex temporal evolutions and spatial distributions introduce a level of complexity that traditional VQ methods are not equipped to handle. Specifically, the unsatisfactory results of traditional VQ methods are caused by two reasons:

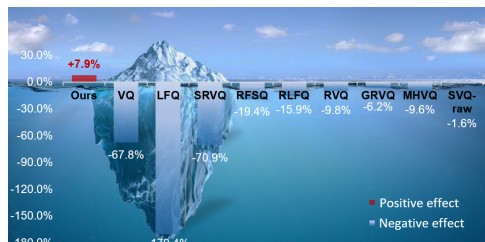

Figure 1: **Limitations of VQ in spatio-temporal forecasting:** An experiment study evaluating mean squared error (MSE) improvement percentage on the WeatherBench-S temperature dataset.

**Inaccurate model optimization caused by non-differentiability.** The discrete nature of the quantization step prevents gradients from being directly passed through this operation. VQ methods typically employ the straight-through (or stop-gradient) estimator, as described in VQVAE van den Oord et al. (2017). This estimator approximates the gradient by copying gradients from the quantized outputs to the input vectors, introducing errors in the optimization process.

**Limited representation power of hard-VQ.** VQ methods typically assign each input vector to a single nearest codebook vector, which limits the modeling of detailed spatio-temporal dynamics required for forecasting.

In response to these limitations, this work presents Differentiable Sparse Soft-Vector Quantization (SVQ), a novel technique designed to strike a balance between noise reduction and detail preservation for spatio-temporal forecasting tasks. We solve the aforementioned challenges by:

- Introducing a differentiable VQ that simplifies gradient computations. This is achieved by approximating sparse regression with an MLP layer, coupled with a codebook, retaining the differentiability crucial for modern deep learning pipelines. A two-layer MLP generates regression coefficients through nonlinear projections of input vectors. The quantized outputs are derived from the dot product of these coefficients and the codebook matrix. Since the coefficients are generated from input vectors, gradients can flow directly from the quantized outputs to the input vectors. This straightforward yet effective approach not only enables differentiable VQ, enhancing accuracy, but also addressing the computational challenges often associated with sparse regression in VQ.

- Using Soft-VQ with sparse regression to combine vectors from a large codebook. As shown in Figure 2, SVQ innovatively integrates sparse regression and allows for the allocation of input vectors with multiple codebook vectors. This significantly enhances the model's ability to capture intricate patterns and effectively filter out noise. Compared to hard-VQ, SVQ exhibits a more uniform distribution of codebook vectors, indicating that SVQ is able to preserve more diverse and fine-grained information from the original inputs. Our empirical studies reveal that SVQ possesses intriguing properties, such as effectively utilizing a completely frozen, randomly initialized codebook without sacrificing performance, thereby significantly reducing learning parameters and showcasing its efficiency and robustness.

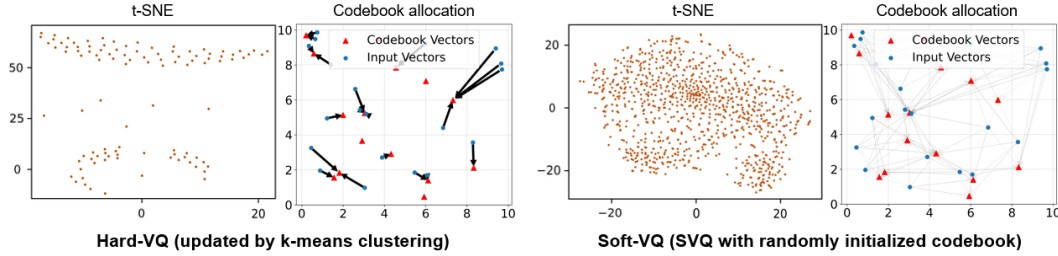

Figure 2: **Hard-VQ vs Soft-VQ:** Using t-SNE, we visualize the codebook vectors on WeatherBench-S temperature dataset with the same codebook size (1024). We provide examples of codebook allocation, where hard-VQ typically assigns each input vector to a single codebook vector as described in VQ-VAE van den Oord et al. (2017), whereas soft-VQ assigns each input vector to multiple codebook vectors.

Through rigorous testing on a variety of real-world datasets, **SVQ has proven to be the first VQ method to achieve significant enhancements in spatio-temporal forecasting tasks.** Notably, SVQ

surpassed the leading model in the WeatherBench-S temperature forecasting benchmark by 7.9%. In video prediction tasks—Human3.6M, KTH, and KittiCaltech, SVQ systematically lowered the Mean Absolute Error (MAE) by 9.4%, while also marking a significant improvement in perceptual quality, as indicated by a 17.3% reduction in the LPIPS score. These results underscore SVQ's remarkable capability across a wide range of spatio-temporal forecasting tasks.

## 2 RELATED WORK

Due to space limitations, here we provide a brief overview of vector quantization, the lineage of sparse coding techniques, and the latest developments in spatio-temporal forecasting algorithms. An extensive review can be found in the Appendix B.

**Vector Quantization and Sparse Coding.** Instead of using continuous latent, VQ-VAE van den Oord et al. (2017), a seminal work, incorporates vector quantization to learn discrete latent representations, typically assigning each vector to the nearest code in a codebook. Subsequent enhancements include Residual VQ Zeghidour et al. (2022), which quantizes the residuals recursively, and Multi-headed VQ Mama et al. (2021b), which adopts multiple heads for each vector. While these methods are effective, they often rely on a relatively small number of codes to represent the original vectors. To address this, SCVAE Xiao et al. (2023) employs sparse coding, allowing vectors to be represented through sparse linear combinations of multiple codes, and achieves end-to-end training via the Learnable Iterative Shrinkage Thresholding Algorithm (LISTA) Gregor & LeCun (2010). However, a significant drawback of the sparse coding method using LISTA (referred to as SVQ-raw here) is its high computational complexity, which scales quadratically with codebook size.

Building on these insights and limitations, our work proposes a soft-VQ method applicable to spatio-temporal forecasting tasks. Although it is closely related to a simultaneous research work Tschannen et al. (2023), which employs an infinite cookbook with a linear layer for continuous vector quantization applied in image generation, our approach is largely inspired by sparse regression, as clearly evidenced by our analysis. Specifically, our work focuses on the challenges arising from spatio-temporal forecasting, providing a strong theoretical foundation and effectively addressing the challenges.

**Spatio-Temporal Forecasting Models.** Recent advancements in spatio-temporal forecasting have highlighted a shift from recurrent to non-recurrent frameworks. Despite the forecasting capabilities of models like ConvLSTM SHI et al. (2015), PredRNN Wang et al. (2017), and PredRNNV2Wang et al. (2022), this shift is largely due to the high computational demands of sequential processing in recurrent models. Non-recurrent models, such as MMVP Zhong et al. (2023) and the SimVP family Gao et al. (2022); Tan et al. (2022), have become benchmarks in video prediction by decoupling spatial and temporal learning through an efficient encoder-translator-decoder structure. This transition is further enhanced by innovative features like visual attention in TAU Tan et al. (2023a) and MetaFormers in OpenSTL Tan et al. (2023b), showcasing the continuous improvements towards more effective forecasting solutions. Our proposed method is designed for seamless integration as a plugin with the majority of these spatio-temporal forecasting models.

## 3 DIFFERENTIABLE SPARSE SOFT-VECTOR QUANTIZATION (SVQ)

In this section, we will first outline the mathematical foundation of sparse soft-vector quantization, followed by a detailed implementation within a spatio-temporal forecasting model. Our proposed method effectively addresses the **optimization problem** through differentiation, and the subsequent theoretical analysis of cookbook utilization demonstrates its substantial **representational capacity**.

### 3.1 VECTOR QUANTIZATION BY SPARSE REGRESSION

Let $\{z_i \in \mathbb{R}^d\}_{i=1}^m$ be the set of codes. A typical vector quantization method assigns a data point $x \in \mathbb{R}^d$ to the nearest code in $\{z_i\}_{i=1}^m$. The main problem with such an approach is that a significant part of the information in $x$ will be lost due to quantization. Sparse regression turns the code assignment problem into an optimization problem

$$w = \underset{w \in \mathbb{R}_+^m}{\arg\min} \frac{1}{2}\left|x - \sum_{i=1}^m w_i z_i\right|^2 + \lambda|w|_1, \tag{1}$$

where $w = (w_1, \ldots, w_m) \in \mathbb{R}^m_+$ is the weight for combining codes $\{z_i\}_{i=1}^m$ to approximate $x$. $\lambda$ refers to the regularization parameter. By introducing $L_1$ regularizer in the optimization problem, we effectively enforce $x$ to be associated with a small number of codes. Compared to classic VQ methods where codes have to be learned through clustering, according to Chiu et al. (2022), it is sufficient to use randomly sampled vectors as codes as long as its number is large enough, thus avoiding the need of computing and adjusting codes. The theoretical guarantee of sparse regression is closely related to the property of subspace clustering, as revealed in Theorem 4.1.

As shown in Figure 3, the obvious downside of sparse regression for VQ is its high computational cost, as it needs to solve the optimization problem in (1) for EVERY data point. Below, we will show that sparse regression can be approximated by a two-layer MLP and a randomly fixed or learnable matrix, making it computationally attractive.

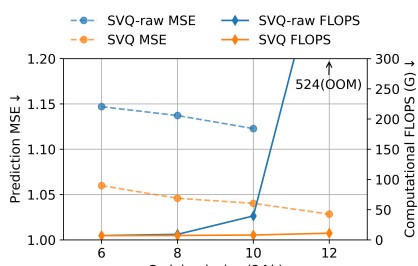

To solve the optimization problem (1), we consider the composite optimization method whose iteration is given as follows

$$w'_{t+1} = w_t - \eta Z^\top (Z w_t - x), \qquad (2)$$
$$[w_{t+1}]_i = \text{sgn}\left([w'_{t+1}]_i\right)\left(\left|[w'_{t+1}]_i\right| - \lambda\eta\right)_+, \qquad (3)$$

where $Z = (z_1, \ldots, z_m)$, $[z]_i$ is the $i$th element of vector $z$, sgn denotes the sign function, and $(a)_+$ outputs $0$ if $a < 0$ and $a$ otherwise. We consider the first step of the iteration where $w_0 = 0$ and have $w = \eta \text{sgn}\left(Z^\top x - \lambda \mathbf{1}\right)\left[Z^\top x - \lambda \mathbf{1}\right]_+ = \eta \text{sgn}\left(Z^T x - \lambda\right)\sigma\left(Z^T x - \lambda\right)$ and the resulting output for $x$ is given as

Figure 3: **Effect of SVQ approximation:** Floating point operations per second (FLOPS) and mean squared error (MSE) on WeatherBench-S temperature dataset with SVQ-raw and SVQ. The computational complexity of SVQ-raw increases quadratically with the size of codebook, making it suffer from out-of-memory (OOM) issue when scaling codebook size up to $2^{12}$.

$$x' = \sum_{i=1}^m w_i z_i = \eta Z \, sgn\left(Z^T x - \lambda\right)\sigma\left(Z^T x - \lambda\right). \qquad (4)$$

By generalizing $\eta Z$ into another matrix $B$, we have output vector $x'$ exactly expressed as a matrix and a two-layer MLP over $x$. We finally note that although it is convenient to form the codebook by randomly sampling vectors, we found empirically that tuning codebook does bring slight additional gains in some cases.

## 3.2 Spatio-Temporal Forecasting Model Enhanced by Quantization

**Architecture of backbone model.** As depicted in Figure 4, SimVP Tan et al. (2022) is employed as the backbone model, which encompasses an encoder for spatial feature extraction, a translator for temporal dependency learning, and a decoder for frame reconstruction. The quantization module is integrated between the encoder and translator. The input data is a 4D tensor $X \in \mathbb{R}^{H*W*T*C}$, representing height ($H$), width ($W$), time step ($T$), and channel ($C$). The encoder En condenses $X$ into downsampled latent representation $\text{En}(X) \in \mathbb{R}^{H'*W'*T*C'}$, maintaining temporal dimensionality while altering spatial and channel dimensions. This latent space, composed of $H' * W' * T$ tokens, each represented by a $C'$-dimensional vector, undergoes vector quantization.

**Quantization module.** The SVQ comprises a two-layer MLP and an extensive codebook. The codebook is a randomly initialized matrix $\mathcal{M} \in \mathbb{R}^{N*C'}$, where $N$ denotes the size of codebook. To achieve automatic selection of codes, a weight matrix $\mathcal{W} \in \mathbb{R}^{H'*W'*T*N}$ is generated via nonlinear projection from the latent representation $\text{En}(X)$. This projection is formally expressed as $\mathcal{W} = \text{MLP}(\text{En}(X))$, wherein the MLP comprises two linear layers and an intermediate ReLU activation function. The quantized output $\mathcal{Q}$ is then obtained by computing the dot product of weight matrix $\mathcal{W}$ and codebook matrix $\mathcal{M}$, a process that can be conceptualized as a selection operation as shown in Figure 4. To encourage sparsity within the generated weight matrix, we apply a Mean Absolute Error (MAE) loss to the output as a surrogate form of regularization.

Figure 4: **Top: Architecture of backbone model and the proposed quantization module.** The encoder, translator, decoder are inherited from SimVP. A quantization module is added between the encoder and translator to effectively ensure a good generalized performance. **Bottom: Quantization process of traditional VQ (Left) and our proposed SVQ (Right)**. In contrast, SVQ select multiple codes (red dots) from a huge codebook (gray dots), and the codebook can be either learnable or frozen.

## 4 EFFICIENT UTILIZE OF COOKBOOK USING SPARSE REGRESSION

To understand the difference between sparse regression-based quantization scheme and clustering-based quantization scheme, we measure the number of codes required to approximate any vector within a unit ball $\mathcal{B}$ with error less than $\delta$. This number is denoted by $T(\mathcal{B}, \delta)$. Intriguingly, as the theorem below reveals, using sparse regression allows $T(\mathcal{B}, \delta)$ to be significantly reduced from $O(1/\delta^d)$ to $O(1/\delta^p)$, where $p \ll d$ for high-dimensional vectors.

**Theorem 4.1.** *For the clustering-based method, $T(\mathcal{B}, \delta)$ is at least $1/\delta^d$. In contrast, for sparse regression, $T(\mathcal{B}, \delta)$ can be formulated as $(4d/\delta)^p$, where*

$$p \geq \max\left(3, \frac{\log(4/\delta)}{\log\log(2d/\varepsilon)}\right), \tag{5}$$

*given that the number of non-zero elements utilized by sparse regression is at least*

$$\frac{4d}{\delta\left(\log C + p\log(4d) - (p+1)\log\delta\right)}. \tag{6}$$

*Proof.* To estimate $T(\mathcal{B}, \delta)$ for the clustering method, we consider the covering number for a unit ball $\mathcal{B}$ which necessitates at least $1/\delta^d$ code vectors to approximate any vector within an acceptable error margin of $\delta$. With $U = (u_1, \ldots, u_m)$ where $u_k \sim \mathcal{N}(0, I_d/m)$, and $g \in \Delta_s$ an $s$-sparse unit vector, we discern:

$$\Pr\left(\|UU^\top - I\|_2 \geq \gamma\right) \leq 2d\exp\left(-\frac{m\gamma^2}{3d}\right), \tag{7}$$

which implies

$$\|UU^\top - I\|_2 \leq \Delta := \sqrt{\frac{d}{m}\log\frac{2d}{\varepsilon}} \tag{8}$$

with probability at least $1 - \varepsilon$. Therefore, $\|g' - g\|_2 \geq (1 + \Delta)^{-1}\|Ug - Ug'\|_2$. Since the $s$-sparse unit vector covering number is bounded by $(Cm/s\delta)^s$, we establish:

$$\left(\frac{Cm}{s\delta}\right)^s \geq \left(1 + \frac{2}{\delta}\right)^d (1 + \Delta)^d, \tag{9}$$

Setting $m = (4d/\delta)^p$ yields

$$(1 + \Delta)^d \leq \exp(d\Delta) \leq e, \tag{10}$$

therefore, $s\log(C'm/\delta) \geq d(2/\delta + \log(1 + \Delta))$, where $C' = Ce$. As long as $s \geq \frac{4d}{\delta(\log C + p\log(4d) - (p+1)\log\delta)}$, it follows that $s\log s \leq 2d/\delta$, affirming that $m \geq (4d/\delta)^p$.

## 5 EXPERIMENTS

We extensively evaluate SVQ on a wide range of real-world spatio-temporal forecasting datasets under the unified framework of OpenSTL Tan et al. (2023b). Given that SimVP holds leading performance across almost all benchmarks, it serves as our primary baseline.

**Dataset.** We conduct extensive experiments on five real-world spatio-temporal forecasting tasks, including weather (WeatherBench Rasp et al. (2020)), traffic flow (TaxiBJ Zhang et al. (2017)), human pose dynamics (Human3.6M Ionescu et al. (2014)), driving scenes (KittiCaltech Geiger et al. (2013); Dollár et al. (2009)), and human actions (KTH Action Schüldt et al. (2004)). The above datasets have relatively few channels. As the number of channels increases, it becomes more challenging to apply VQ, requiring a more diverse codebook. Therefore, to validate the performance on high-dimensional data, additional experiments were conducted using the WeatherBench dataset in a High-dimensional Multi-Variable (HMV) setting, which includes a total of 110 meteorological factors. Details about datasets are provided in Appendix A.1.

**Experimental details.** During deployment, we found SVQ to be quite robust to codebook size, as its performance remains consistently strong when using a sufficiently large codebook. Therefore, we fix the codebook size at 10,000 for WeatherBench, TaxiBJ, and Human3.6M datasets, and at 6,000 for KittiCaltech and KTH datasets. The hidden dimension of nonlinear projection layer is fixed at 128. Experiments are conducted on either 1 or 4 NVIDIA V100 32GB GPUs, with a total batch size of 16. More details about backbone architectures, VQ parameters, computational costs, and metrics are described in Appendix A.2, A.3, D.1, and A.5, respectively.

### 5.1 BENCHMARKS ON VARIOUS FORECASTING TASKS

We explore both fixed (frozen) and learnable versions of SVQ on various forecasting tasks. Interestingly, our findings reveal that with a large codebook size, the performance of a frozen, randomly-initialized codebook is on par with that of a carefully learned codebook. This observation aligns with our intuition: when allowed to choose a very large number of representative vectors to form a codebook, a random choice is often as good as the one that is carefully chosen, which has already been studied in the column subset selection problem in matrix theory Drineas et al. (2008); Deshpande & Rademacher (2010). The comparison baselines consist of two categories: 1) Non-recurrent models including SimVP Tan et al. (2022) and TAU Tan et al. (2023a); 2) Recurrent-based models including ConvLSTM SHI et al. (2015), PredNet Lotter et al. (2017), PredRNN Wang et al. (2017), PredRNN++ Wang et al. (2018), MIM Wang et al. (2019b), E3D-LSTM Wang et al. (2019a), PhyDNet Guen & Thome (2020), MAU Chang et al. (2021), PredRNNv2 Wang et al. (2022), and DMVFN Hu et al. (2023). Baseline results are copied from the original OpenSTL paper Tan et al. (2023b). To preclude ambiguity, we select the best MetaFormer of SimVP for each dataset, detailed in Appendix A.2.

The benchmark results of WeatherBench and three video prediction datasets (Human3.6M, KTH, and KittiCaltech) are presented in Tables 1 and 2, respectively. Due to page limit, results of WeatherBench-HMV and TaxiBJ datasets are provided in Appendix D.8 and D.7. These datasets have different characteristics. WeatherBench and TaxiBJ are macro forecasting tasks with low-frequency collection (30min or 1-6h). Human3.6M features subtle, low-frequency frame differences. KittiCaltech is challenging due to rapidly changing backgrounds and limited training data. The KTH dataset tests long-horizon forecasting, requiring the prediction of 20 future frames from 10 observed frames.

However, despite the distinct characteristics among datasets, a common thread is the need for improved noise reduction coupled with enhanced representational capabilities, which can universally benefit their respective forecasting tasks. Notably, the SimVP+SVQ model achieves either the best or comparable performance across all datasets. For instance, on the WeatherBench-S temperature dataset, SVQ significantly improves the best baseline by **7.9%** ($1.105 \rightarrow 1.018$). On these three popular video prediction tasks, SVQ not only delivers a reduction in forecasting errors (average **9.4%** decrease in MAE), but also significantly improves subjective image quality (average **17.3%** decrease in LPIPS). On the WeatherBench-HMV dataset, SVQ continues to demonstrate a reduction in MAEs in 110 channels, with an average of **8.9%**. The results affirm that SVQ maintains good performance when applied to high-dimensional datasets. Additional visualizations of forecasting samples can be found in Appendix F.

Table 1: **WeatherBench results:** Performance comparison for SVQ module and baseline models on Weather-Bench. WeatherBench-S is single-variable, one-hour interval forecasting setup trained on data from 2010-2015, validated on 2016, and tested on 2017-2018. WeatherBench-M targets broader application, which is multi-variable, six-hour interval forecasting setup trained on data from 1979-2015, validated on 2016, and tested on 2017-2018. The best and the second best results are highlighted by **bold** and underlined.

| Dataset | Variable | Temperature | | Humidity | | Wind Component | | Total Cloud Cover | |
| --- | --- | --- | --- | --- | --- | --- | --- | --- | --- |
| | Model | MSE↓ | MAE↓ | MSE↓ | MAE↓ | MSE↓ | MAE↓ | MSE↓ | MAE↓ |
| WeatherBench-S | ConvLSTMSHI et al. (2015) | 1.521 | 0.7949 | 35.146 | 4.012 | 1.8976 | 0.9215 | 0.0494 | 0.1542 |
| | E3D-LSTMWang et al. (2019a) | 1.592 | 0.8059 | 36.534 | 4.100 | 2.4111 | 1.0342 | 0.0573 | 0.1529 |
| | PredRNNWang et al. (2017) | 1.331 | 0.7246 | 37.611 | 4.096 | 1.8810 | 0.9068 | 0.0550 | 0.1588 |
| | MIMWang et al. (2019b) | 1.784 | 0.8716 | 36.534 | 4.100 | 3.1399 | 1.1837 | 0.0573 | 0.1529 |
| | MAUChang et al. (2021) | 1.251 | 0.7036 | 34.529 | 4.004 | 1.9001 | 0.9194 | 0.0496 | 0.1516 |
| | PredRNN++Wang et al. (2018) | 1.634 | 0.7883 | 35.146 | 4.012 | 1.8727 | 0.9019 | 0.0547 | 0.1543 |
| | PredRNN.V2Wang et al. (2022) | 1.545 | 0.7986 | 36.508 | 4.087 | 2.0072 | 0.9413 | 0.0505 | 0.1587 |
| | TAUTan et al. (2023a) | 1.162 | 0.6707 | 31.831 | 3.818 | 1.5925 | 0.8426 | 0.0472 | 0.1460 |
| | SimVP (w/o VQ)Tan et al. (2022) | 1.105 | 0.6567 | 31.332 | 3.776 | 1.4996 | 0.8145 | 0.0466 | 0.1469 |
| | **SimVP+SVQ (Frozen codebook)** | 1.023 | 0.6131 | 30.863 | 3.661 | 1.4337 | 0.7861 | **0.0456** | **0.1456** |
| | **SimVP+SVQ (Learnable codebook)** | **1.018** | **0.6109** | **30.611** | **3.657** | **1.4186** | **0.7858** | 0.0458 | 0.1463 |
| | **Improvement** | ↑7.9% | ↑7.0% | ↑2.3% | ↑3.2% | ↑5.4% | ↑3.5% | ↑2.1% | ↑0.9% |
| | Variable | Temperature | | Humidity | | Wind U Component | | Wind V Component | |
| WeatherBench-M | ConvLSTMSHI et al. (2015) | 6.303 | 1.7695 | 368.15 | 13.490 | 30.002 | 3.8923 | 30.789 | 3.8238 |
| | PredRNNWang et al. (2017) | 5.596 | 1.6411 | 354.57 | 13.169 | 27.484 | 3.6776 | 28.973 | 3.6617 |
| | MIMWang et al. (2019b) | 7.515 | 1.9650 | 408.24 | 14.658 | 35.586 | 4.2842 | 36.464 | 4.2066 |
| | MAUChang et al. (2021) | 5.628 | 1.6810 | 363.36 | 13.503 | 27.582 | 3.7409 | 27.929 | 3.6700 |
| | PredRNN++Wang et al. (2018) | 5.647 | 1.6433 | 363.15 | 13.246 | 28.396 | 3.7322 | 29.872 | 3.7067 |
| | PredRNN.V2Wang et al. (2022) | 6.307 | 1.7770 | 368.52 | 13.594 | 29.833 | 3.8870 | 31.119 | 3.8406 |
| | TAUTan et al. (2023a) | 4.904 | 1.5341 | 342.63 | 12.801 | 24.719 | 3.5060 | 25.456 | 3.4723 |
| | SimVP (w/o VQ)Tan et al. (2022) | 4.833 | 1.5246 | 340.06 | 12.738 | 24.535 | 3.4882 | 25.232 | 3.4509 |
| | **SimVP+SVQ (Frozen codebook)** | **4.427** | **1.4160** | 360.15 | **12.445** | 23.915 | 3.4078 | **24.968** | 3.4117 |
| | **SimVP+SVQ (Learnable codebook)** | 4.433 | 1.4164 | 360.53 | 12.449 | **23.908** | **3.4060** | 24.983 | **3.4095** |
| | **Improvement** | ↑8.4% | ↑7.1% | ↓5.9% | ↑2.3% | ↑2.6% | ↑2.4% | ↑1.0% | ↑1.2% |

Table 2: **Video prediction results:** Performance comparison for SVQ module and baseline models on Human3.6M, KTH, and KittiCaltech. The best and the second best results are highlighted by **bold** and underlined.

| Dataset | Human3.6M | | | | KittiCaltech | | | | KTH | | | |
| --- | --- | --- | --- | --- | --- | --- | --- | --- | --- | --- | --- | --- |
| Metric | MAE↓ | SSIM↑ | PSNR↑ | LPIPS↓ | MAE↓ | SSIM↑ | PSNR↑ | LPIPS↓ | MAE↓ | SSIM↑ | PSNR↑ | LPIPS↓ |
| ConvLSTMSHI et al. (2015) | 1583.3 | 0.9813 | 33.40 | 0.03557 | 1583.3 | 0.9345 | 27.46 | 0.08575 | 445.5 | 0.8977 | 26.99 | 0.26686 |
| E3D-LSTMWang et al. (2019a) | 1442.5 | 0.9803 | 32.52 | 0.04133 | 1946.2 | 0.9047 | 25.45 | 0.12602 | 892.7 | 0.8153 | 21.78 | 0.48358 |
| PredNetLotter et al. (2017) | 1625.3 | 0.9786 | 31.76 | 0.03264 | 1568.9 | 0.9286 | 27.21 | 0.11289 | 783.1 | 0.8094 | 22.45 | 0.32159 |
| PhyDNetGuen & Thome (2020) | 1614.7 | 0.9804 | **39.84** | 0.03709 | 2754.8 | 0.8615 | 23.26 | 0.32194 | 765.6 | 0.8322 | 23.41 | 0.50155 |
| MAUChang et al. (2021) | 1577.0 | 0.9812 | 33.33 | 0.03561 | 1800.4 | 0.9176 | 26.14 | 0.09673 | 471.2 | 0.8945 | 26.73 | 0.25442 |
| MIMWang et al. (2019b) | 1467.1 | 0.9829 | 33.97 | 0.03338 | 1464.0 | 0.9409 | 28.10 | 0.06353 | 380.8 | 0.9025 | 27.78 | 0.18808 |
| PredRNNWang et al. (2017) | 1458.3 | 0.9831 | 33.94 | 0.03245 | 1525.5 | 0.9374 | 27.81 | 0.07395 | 380.6 | 0.9097 | 27.95 | 0.21892 |
| PredRNN++Wang et al. (2018) | 1452.2 | 0.9832 | 34.02 | 0.03196 | 1453.2 | 0.9433 | 28.02 | 0.13210 | 370.4 | 0.9124 | 28.13 | 0.19871 |
| PredRNN.V2Wang et al. (2022) | 1484.7 | 0.9827 | 33.84 | 0.03334 | 1610.5 | 0.9330 | 27.12 | 0.08920 | 368.8 | 0.9099 | 28.01 | 0.21478 |
| TAUTan et al. (2023a) | 1390.7 | 0.9839 | 34.03 | 0.02783 | 1507.8 | 0.9456 | 27.83 | 0.05494 | 421.7 | 0.9086 | 27.10 | 0.22856 |
| DMVFN Hu et al. (2023) | - | - | - | - | 1531.1 | 0.9314 | 26.95 | **0.04942** | 413.2 | 0.8976 | 26.65 | **0.12842** |
| SimVP (w/o VQ)Tan et al. (2022) | 1441.0 | 0.9834 | 34.08 | 0.03224 | 1507.7 | 0.9453 | 27.89 | 0.05740 | 397.1 | 0.9065 | 27.46 | 0.26496 |
| **SimVP+SVQ (Frozen codebook)** | **1264.9** | **0.9851** | 34.07 | 0.02380 | 1408.6 | **0.9469** | 28.10 | 0.05535 | 364.6 | 0.9109 | 27.28 | 0.20988 |
| **SimVP+SVQ (Learnable codebook)** | 1265.1 | **0.9851** | 34.06 | **0.02367** | 1414.9 | 0.9458 | 28.10 | 0.05776 | **360.2** | 0.9116 | 27.37 | 0.20658 |
| **Improvement** | ↑12.2% | ↑0.2% | ↓0.0% | ↑26.2% | ↑6.6% | ↑0.2% | ↑0.8% | ↑3.6% | ↑9.3% | ↑0.6% | ↓0.3% | ↑22.0% |

## 5.2 BOOSTING PERFORMANCE AS A VERSATILE PLUG-IN

In this section, SVQ serves as a versatile plug-in module applicable to various MetaFormers Yu et al. (2022a). The adopted MetaFormers include three types. 1) CNN-based: SimVPv1(IncepU) Gao et al. (2022), SimVPv2(gSTA) Tan et al. (2022), ConvMixer Trockman & Kolter (2023), ConvNeXt Liu et al. (2022b), HorNet Rao et al. (2022), and MogaNet Li et al. (2022). 2) Transformer-based: ViT Dosovitskiy et al. (2021), Swin Transformer Liu et al. (2021), Uniformer Li et al. (2023), Poolformer Yu et al. (2022b), and VAN Guo et al. (2023). 3) MLP-based: MLPMixer Tolstikhin et al. (2021). We conduct experiments on WeatherBench-S temperature dataset because it is lightweight and fast for training.

Table 3: **Boosting performance:** The effect of SVQ for various MetaFormers on WeatherBench-S temperature dataset.

| MetaFormer | MSE | | MAE | |
| --- | --- | --- | --- | --- |
| | w/o SVQ | w SVQ | w/o SVQ | w SVQ |
| SimVPv1(IncepU)Gao et al. (2022) | 1.238 | **1.216** | 0.7037 | **0.6831** |
| SimVPv2(gSTA)Tan et al. (2022) | 1.105 | **1.018** | 0.6567 | **0.6109** |
| ConvMixerTrockman & Kolter (2023) | 1.267 | **1.257** | 0.7073 | **0.6780** |
| ConvNeXtLiu et al. (2022b) | 1.277 | **1.159** | 0.7220 | **0.6568** |
| HorNetRao et al. (2022) | 1.201 | **1.130** | 0.6906 | **0.6472** |
| MogaNetLi et al. (2022) | 1.152 | **1.067** | 0.6665 | **0.6271** |
| ViTDosovitskiy et al. (2021) | 1.146 | **1.111** | 0.6712 | **0.6375** |
| SwinLiu et al. (2021) | 1.143 | **1.088** | 0.6735 | **0.6320** |
| UniformerLi et al. (2023) | 1.204 | **1.110** | 0.6885 | **0.6400** |
| PoolformerYu et al. (2022b) | 1.156 | **1.097** | 0.6715 | **0.6297** |
| VANGuo et al. (2023) | 1.150 | **1.083** | 0.6803 | **0.6342** |
| MLP-MixerTolstikhin et al. (2021) | 1.255 | **1.120** | 0.7011 | **0.6455** |
| **Average improvement** | ↑4.8% | | ↑6.0% | |

As shown in Table 3, SVQ consistently improves the performance across all MetaFormers, showcasing its universality across diverse backbone types. We observe an average reduction in MSE and MAE by **4.8%** and **6.0%**, respectively. In detail, SVQ leads to an average MSE reduction of 4.1% for CNN-based backbones, 5.1% for transformer-based, and 10.7% for MLP-based. The more pronounced enhancement in transformer-based and MLP-based models indicates that our approach is especially effective with architectures that prioritize global interactions. Notably, SimVPv2(gSTA) is the best backbone, while our SVQ further improves it by 7.9%. These findings also aligns with our motivation that mitigating noise in the learning process significantly benefits spatio-temporal forecasting, irrespective of model architecture. By integrating SVQ to constrain the diversity of predicted patterns and cut out noise, researchers can focus on crafting high-quality and general base models.

## 5.3 DELICATE BALANCE BETWEEN DETAIL PRESERVATION AND NOISE REDUCTION

To study the role of VQ in spatio-temporal forecasting, we evaluated several cutting-edge VQ methods akin to the SVQ framework, implemented as plug-in modules alongside the backbone forecasting model. Table 4 shows that SVQ significantly improves forecasting as a plug-in, whereas other VQ methods result in increased prediction errors. Enhanced detail retention within the representational capacity is associated with lower forecasting errors. Classic VQ methods suffer from notable information losses, as evidenced by a higher MSE of 1.854. In contrast, both residual VQ and grouped residual VQ outperform traditional VQ with lower

Table 4: **Comparison of vector quantization methods:** All methods share identical backbone, with the recommended setting in Appendix A.3. The results better than baseline are highlighted in **bold**.

| Method | MSE↓ | MAE↓ |
|---|---|---|
| Baseline (SimVP w/o VQ) | 1.105 | 0.6567 |
| VQ van den Oord et al. (2017) | 1.854 | 0.8963 |
| Residual VQ (RVQ) Zeghidour et al. (2022) | 1.213 | 0.6910 |
| Grouped Residual VQ (GRVQ) Yang et al. (2023) | 1.174 | 0.6747 |
| Multi-headed VQ (MHVQ) Mama et al. (2021b) | 1.211 | 0.6994 |
| Stochastic Residual VQ (SRVQ) Lee et al. (2022) | 1.888 | 0.9237 |
| Residual Finite Scalar Quantization (RFSQ) Mama et al. (2021a) | 1.319 | 0.7505 |
| Lookup Free Quantization (LFQ) Yu et al. (2023a) | 2.988 | 1.1103 |
| Residual LFQ (RLFQ) Yu et al. (2023a) | 1.281 | 0.7281 |
| SVQ-raw Xiao et al. (2023) | 1.123 | **0.6456** |
| SVQ | **1.018** | **0.6109** |

MSEs of 1.213 and 1.174, respectively, affirming their ability to preserve intricate details due to recursive quantization.

It is commonly understood that the codebook size in clustering-based VQ is critical: larger codebooks capture more details, whereas smaller ones enhance noise reduction. To explore this trade-off, we compared how the codebook size influences the prediction MSE in Grouped Residual VQ (GRVQ) and SVQ. As Figure 5 indicates, the MSE of GRVQ initially decreases but increases with overly large codebooks, echoing findings from Yu et al. (2023b) that an excessively large codebook may degrade image generation performance. This underscores the necessity for dataset-specific tuning in clustering-based VQ approaches. In contrast, SVQ naturally achieves a balance

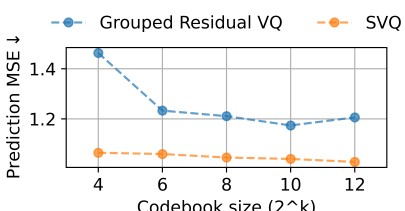

Figure 5: **Predition MSE curves** on WeatherBench-S temperature dataset with Grouped Residual VQ (GRVQ) and SVQ.

between preserving detail and reducing noise through sparse regression, eliminating the need for extensive fine-tuning. The codebook size in SVQ exhibits a low-maintenance profile: using a default large codebook can produce robust results without extensive tuning. We are not suggesting that our SVQ outperforms other VQ methods in general image generation tasks, as it is beyond the scope of our current objective. Rather, we emphasize SVQ's effectiveness as a noise reduction tool that directly enhances real-world spatio-temporal forecasting tasks, while the application to general image generation remains a topic for future exploration.

## 5.4 TRAIN STABILITY ISSUE

Although it is feasible to place the quantization module either before or after the translator, we found that for post-translator placement, the traditional VQ method van den Oord et al. (2017) suffers pronounced instability and codebook collapse issues, as shown in Figure 6. It is essential to highlight that the backbones without VQ maintain their MSE within the acceptable range of approximately 1 to

2 (refer to Table 3). Yet, integrating traditional VQ causes a substantial rise in MSE values, exceeding 10 for different backbones—a level considered excessively high. We hypothesize that this instability is attributed to the non-differentiability of the straight-through estimator, which introduces errors into the gradient flow for preceding modules. In contrast, our SVQ module never encounters such issues and remains highly stable throughout training. To maintain the integrity of all VQ methods, we opt for the pre-translator design in our main experiments, wherein quantization is executed preceding the translator module. The difference between two designs is detailed in Appendix D.3.

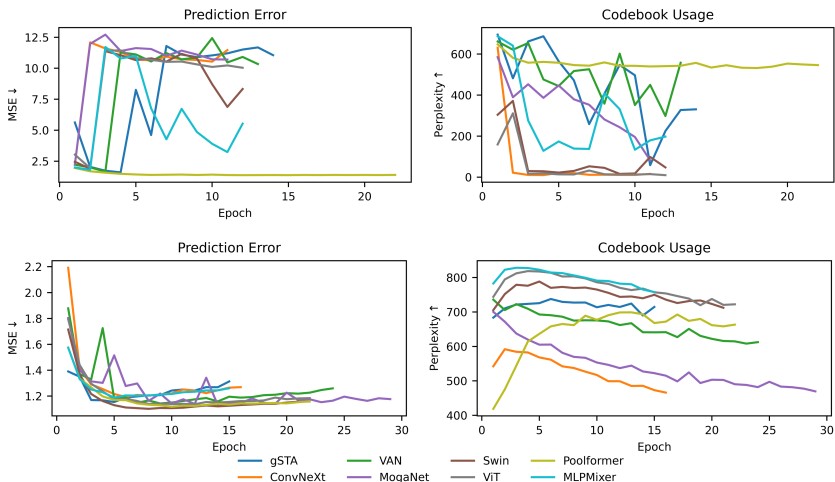

Figure 6: **VQ (Top) and SVQ (Bottom) training curves:** We perform post-translator quantization on various backbones, with the same codebook size (1024), employing early stopping (patience of 10) on the WeatherBench-S temperature dataset. Perplexity for SVQ is averaged over different $\theta$ values, detailed in Appendix A.4.

## 5.5 ABLATION STUDY

We conduct a series of ablation studies on WeatherBench-S temperature dataset to understand the contribution of important designs based on the default setting: SVQ with a codebook size of 10,000, learnable codebook, and MAE loss. An additional ablation study on the frozen module is provided in Appendix D.5.

**Self-learned sparse regression structure.** The original SimVP model adopts MSE as prediction loss. We individually replace it with MAE loss and add the SVQ module. As shown in Table 5, the joint use of SVQ and MAE loss is crucial for significantly improving the model's performance. We suggest that the sparsity of the weight matrix $\mathcal{W}$ impacts vector representation learning and use kurtosis to quantify this after normalizing $\mathcal{W}$. Figure 7 demonstrates that both a learnable codebook and MAE loss contribute to increased sparsity. Analyzing four codebook initialization methods in both learnable and fixed settings (Table 6), we find that a learnable codebook promotes sparsity irrespective of the initialization, indicating that sparsity is a self-learned property that enhances intermediate representation learning.

Table 5: **Ablation of SVQ Compoments.**

| Module | MSE↓ | MAE↓ |
|---|---|---|
| SimVP (MSE loss) | 1.105 | 0.6567 |
| SimVP (MAE loss) | 1.126 | 0.6509 |
| SimVP+SVQ (Learnable, MSE loss) | 1.099 | 0.6527 |
| SimVP+SVQ (Learnable, MAE loss) | **1.018** | **0.6109** |

Table 6: **Ablation of Codebook Initianlization.**

| Initialization | Learnability | MSE↓ | MAE↓ | Kurtosis |
|---|---|---|---|---|
| kaiming uniform | Frozen | 1.023 | 0.6131 | 1.596 |
| | Learnable | **1.018** | **0.6109** | 7.213 |
| sparse(sparsity=0.9) | Frozen | 1.050 | 0.6183 | 4.165 |
| | Learnable | 1.034 | 0.6160 | 41.558 |
| trunc normal | Frozen | 1.049 | 0.6166 | 1.582 |
| | Learnable | 1.031 | 0.6161 | 4.236 |
| orthogonal | Frozen | 1.034 | 0.6170 | 1.561 |
| | Learnable | 1.030 | 0.6131 | 35.774 |

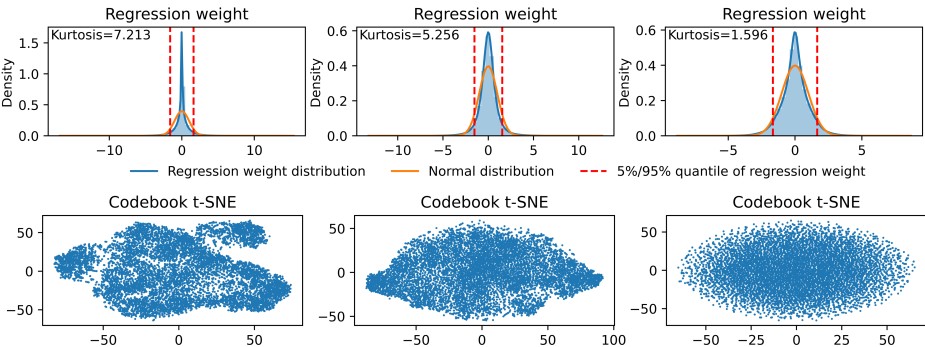

Figure 7: **Distribution of regression weight** $\mathcal{W}$ **and codebook** $\mathcal{M}$**:** Higher kurtosis represents more compact and concentrate distribution near zero, as well as sparser regression weights. Left: Learnable SVQ with MAE loss. Middle: Learnable SVQ with MSE loss. Right: Frozen SVQ with MAE loss. Learnable setting and MAE loss encourage sparser weights and a more structured codebook.

**Codebook size and learnability.** Table 7 compares the effects of codebook size—both learnable and frozen. Results show that increasing codebook size consistently enhances performance. However, when the size reaches 10,000, the performance gap between frozen and learnable codebooks narrows to just 0.5% (1.023 → 1.018). A larger codebook provides comprehensive coverage of the latent space through random codes, minimizing the need for meticulous learning. Consequently, models with randomly initialized codebooks perform similarly to those with learned ones. Additionally, our optimized SVQ structure outperforms alternative designs, including single-layer, bucket-shaped, and post-ReLU variants, while keeping a similar parameter count.

Table 7: **Ablation of Model Structure.**

| Learnability | Projection dim | Codebook size | MSE↓ | MAE↓ |
|---|---|---|---|---|
| Frozen | 128 | 10 | 1.070 | 0.6227 |
| | 128 | 1000 | 1.044 | 0.6198 |
| | 128 | 10000 | 1.023 | 0.6131 |
| Learnable | 128 | 10 | 1.060 | 0.6194 |
| | 128 | 1000 | 1.048 | 0.6182 |
| | 128 | 10000 | **1.018** | **0.6109** |
| | 1280 (Bucket-shape) | 1280 | 1.035 | 0.6149 |
| | None (One-layer) | 10000 | 1.043 | 0.6144 |
| | 128 (Post-ReLU) | 10000 | 1.032 | 0.6136 |

### 5.6 ROBUSTNESS TO NOISE, ERROR-BAR, CONVERGENCE BEHAVIOUR, VISUALIZATIONS OF PREDICTIONS AND LATENT VECTORS, AND HIGH-DIMENSIONAL BENCHMARK RESULTS.

We conducted additional experiments by introducing artificial noise to the training data, confirming that our method effectively mitigates noise by constraining latent patterns through quantization, as detailed in Appendix D.2. The statistical significance of the error bars is provided in Appendix D.6. We further analyzed the convergence behavior of SVQ and traditional VQ in Appendix C. Additionally, we included supplementary experiments to understand the impact of SVQ on latent representation and to compare various VQ methods in Appendices E, and D.4, respectively. A benchmark experiment for high-dimensional spatio-temporal forecasting is also included in Appendix D.8.

## 6 CONCLUSIONS

In this work, we present Differentiable Sparse Soft-Vector Quantization (SVQ), a concise yet effective method for spatio-temporal forecasting enhancement. Unlike other state-of-the-art VQ methods, this is the first approach that demonstrates a boosting effect in spatio-temporal forecasting tasks.SVQ elegantly tackles the inaccuracies in the optimization problem arising from non-differentiability and the restricted representational capabilities associated with hard-VQ.Tested across diverse benchmarks, from weather to traffic and video prediction, SVQ consistently outperforms pure baseline methods, setting new performance standards without complex priors. Its differentiability and seamless integration with baseline models highlight SVQ as a significant advancement for efficient and effective spatio-temporal forecasting.

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

# Supplemental Materials

The supplementary material for our work *Does Vector Quantization Fail in Spatio-Temporal Forecasting? Exploring a Differentiable Sparse Soft-Vector Quantization Approach* is organized as follows: Appendix A provides implementation details of SimVP model and VQ methods. Appendix B gives an extensive review of related work. Appendix C analyzes the convergence behaviour of SVQ and traditional VQ. Appendix D present extended quantitative results. Appendix E delves deeper into the effect of SVQ on latent representation. Finally, Appendix F shows additional qualitative results of forecasting samples and errors.

## A   IMPLEMENTATION DETAILS

### A.1   DATASET DETAILS

WeatherBench Rasp et al. (2020) and TaxiBJ Zhang et al. (2017) are two macro forecasting tasks collected at low frequencies (30min or 1-6h). Human3.6M Ionescu et al. (2014), KittiCaltech Geiger et al. (2013); Dollár et al. (2009), and KTH Action Schüldt et al. (2004) are three popular video prediction tasks. A summary of dataset statistics is provided in Table 8.

Table 8: The detailed statistics of benchmark datasets.

| Dataset | Size | | Seq. Len. | | Img. Shape | Interval |
|---|---|---|---|---|---|---|
| | train | test | in | out | $H \times W \times C$ | |
| WeatherBench-S | 2,167 | 706 | 12 | 12 | $32 \times 64 \times 1$ | 1 hour |
| WeatherBench-M | 54,019 | 2,883 | 4 | 4 | $32 \times 64 \times 4$ | 6 hour |
| TaxiBJ | 20,461 | 500 | 4 | 4 | $32 \times 32 \times 2$ | 30 min |
| KittiCaltech | 3,160 | 3,095 | 10 | 1 | $128 \times 160 \times 3$ | Frame |
| Human3.6M | 73,404 | 8,582 | 4 | 4 | $256 \times 256 \times 3$ | Frame |
| KTH Action | 4,940 | 3,030 | 10 | 20 | $128 \times 128 \times 1$ | Frame |
| WeatherBench-HMV | 52,559 | 2,883 | 4 | 4 | $32 \times 64 \times 110$ | 6 hour |

### A.2   ARCHITECTURE CONFIGURATION OF SIMVP

Table 9 reports the architectures of SimVP on all datasets. We select the best MetaFormer to replace the translator module based on OpenSTL benchmarks[1][2][3]. The parameters remain unchanged, following the original configurations. It is noteworthy that due to reproducibility issues of ConvNeXt on the TaxiBJ dataset, we have opted to utilize gSTA as our backbone model.

Table 9: Detailed configuration of SimVP backbone.

| Dataset | MetaFormer (Translator) | spatio_kernel | hid_S | hid_T | N_T | N_S | drop_path | LR scheduler |
|---|---|---|---|---|---|---|---|---|
| WeatherBench-S temperature | gSTA | enc=3, dec=3 | 32 | 256 | 8 | 2 | 0.1 | cosine |
| WeatherBench-S humidity | Swin | enc=3, dec=3 | 32 | 256 | 8 | 2 | 0.2 | cosine |
| WeatherBench-S wind component | Swin | enc=3, dec=3 | 32 | 256 | 8 | 2 | 0.2 | cosine |
| WeatherBench-S total cloud cover | gSTA | enc=3, dec=3 | 32 | 256 | 8 | 2 | 0.1 | cosine |
| WeatherBench-M | MogaNet | enc=3, dec=3 | 32 | 256 | 8 | 2 | 0.1 | cosine |
| TaxiBJ | gSTA | enc=3, dec=3 | 32 | 256 | 8 | 2 | 0.1 | cosine |
| Human3.6M | gSTA | enc=3, dec=3 | 64 | 512 | 6 | 4 | 0.1 | cosine |
| KTH | IncepU | enc=3, dec=3 | 64 | 256 | 6 | 2 | 0.1 | onecycle |
| KittiCaltech | gSTA | enc=3, dec=3 | 64 | 256 | 6 | 2 | 0.2 | onecycle |
| WeatherBench-HMV | gSTA | enc=3, dec=3 | 32 | 256 | 8 | 2 | 0.1 | cosine |

---

[1] https://openstl.readthedocs.io/en/latest/model_zoos/video_benchmarks.html
[2] https://openstl.readthedocs.io/en/latest/model_zoos/weather_benchmarks.html
[3] https://openstl.readthedocs.io/en/latest/model_zoos/traffic_benchmarks.html

### A.3 Parameters of compared VQ methods

Table 4 presents a comparison of SVQ with several well-known VQ methods, reproduced using source code from the GitHub repository[4]. The parameters were kept consistent with the recommended settings to ensure performance, as detailed in Table 10. It should be noted that we found that increasing the codebook size for previous VQ methods, such as Residual VQ and Multi-headed VQ, led to a considerable increase in GPU memory usage and extended the training time to impractical levels. This issue is one of the reasons these methods recommend adopting a default codebook size of 1024. To ensure fairness, we conducted an extensive experiment for VQ methods using the same codebook size (1024) in Appendix D.4.

Table 10: Parameters of the compared VQ methods.

| Vector quantization method | codebook_size | num_quantizers | groups | heads | shared_codebook | Specific parameters |
|---|---|---|---|---|---|---|
| VQ | 512 | - | - | - | - | - |
| Residual VQ | 1024 | 8 | - | - | ✓ | - |
| Grouped Residual VQ | 1024 | 8 | 2 | - | ✓ | - |
| Multi-headed VQ | 1024 | - | - | 8 | ✓ | - |
| Residual VQ (Stochastic) | 1024 | 8 | - | - | ✓ | stochastic_sample_codes=True |
| Residual Finite Scalar Quantization | - | 8 | - | - | - | levels=[8, 5, 5, 3] |
| Lookup Free Quantization (LFQ) | 8192 | - | - | - | - | entropy_loss_weight=0.1 |
| Residual LFQ | 256 | 8 | - | - | - | - |

### A.4 Evaluation of perplexity

Unlike other VQ methods that rely on a single code, our SVQ generates multiple regression weights to merge several codes. To evaluate its perplexity, we first normalize the regression weights and then convert them into binary form using a threshold set at $\theta$ times the standard deviation, where $\theta$ serves as the threshold value. We utilize two thresholds (2 and 3) to obtain reasonable perplexity.

### A.5 Metrics

Forecasting accuracy is evaluated using mean squared error (MSE) and mean absolute error (MAE), while the image quality of predicted frames is assessed using structural similarity index measure (SSIM) Wang et al. (2004), peak signal-to-noise ratio (PSNR), and learned perceptual image patch similarity (LPIPS) Zhang et al. (2018). The training process is early stopped with a patience of 10, and the models with the minimal loss are saved for subsequent evaluation.

## B Extensive review of related work

### B.1 Recurrent-based forecasting model

The majority of spatio-temporal forecasting models leverage techniques such as Conv2D Xu et al. (2018), Conv3D Wang et al. (2019a), and attention mechanisms Liu et al. (2022a) for spatial modeling. Distinctions among these models primarily arise from how they incorporate temporal information. Recurrent-based models, exemplified by ConvLSTM SHI et al. (2015), have been widely used to capture motion dynamics by iteratively processing multi-frame predictions. Variants like PredRNN Wang et al. (2017) introduce the Spatio-Temporal LSTM (ST-LSTM) unit, integrating spatial appearances and temporal variations within a single memory pool. Further advancements include PredRNN++ Wang et al. (2018) and PredRNNV2 Wang et al. (2022), which deepen the model and expand the receptive field through a cascading LSTM mechanism and a memory decoupling strategy, respectively. MIM Wang et al. (2019b) network utilizes a self-renewed memory module to exploit differential signals by decomposing non-stationary dynamics. PredNet Lotter et al. (2017) improves performance by estimating prediction errors in forward propagation.

### B.2 Non-recurrent forecasting model

Despite the effectiveness of recurrent-based methods, they suffer from high computational cost caused by their inherent unparallelizable architecture. Recent efforts in spatio-temporal forecasting have shifted towards non-recurrent models that decouple the forecasting task from autoregressive processes.

---

[4]https://github.com/lucidrains/vector-quantize-pytorch/tree/master

Notably, SimVPv1 Gao et al. (2022) and SimVPv2 Tan et al. (2022) separate spatial and temporal learning into distinct phases within an encoder-translator-decoder structure, consistently surpassing recurrent counterparts in video prediction tasks. TAU Tan et al. (2023a) refines the architecture by incorporating a visual attention mechanism into the translator. OpenSTL Tan et al. (2023b) further enhances the translator by MetaFormers.

### B.3 VECTOR QUANTIZATION

By partitioning a continuous vector space into a discrete collection of vectors, VQ effectively reduces the data required to characterize a set of values, thereby achieving noise reduction. Following the introduction of VQ-VAE van den Oord et al. (2017), many variants such as VQGAN Esser et al. (2021), Residual VQ Zeghidour et al. (2022), Multi-headed VQ Mama et al. (2021b), Grouped Residual VQ Yang et al. (2023), Finite Scalar Quantization (FSQ) Mama et al. (2021a), and Lookup Free Quantization Yu et al. (2023a) have been developed to enhance the representational capabilities of VQ. For instance, FSQ Mama et al. (2021a) simplifies VQ in generative modeling by discretizing scalar values. However, to the best of our knowledge, VQ has seen limited application in spatio-temporal forecasting, which has inspired our research. Traditional hard-VQ tends to eliminate excessive detail, thus impairing forecasting accuracy. While the sparse coding-based variational autoencoder (SC-VAE) Xiao et al. (2023) incorporates sparse coding into the variational autoencoder framework, its application is primarily targeted at image reconstruction and segmentation tasks. Furthermore, in the main text, our experiments have demonstrated that the implementation of the LISTA algorithm Gregor & LeCun (2010) within SC-VAE leads to out-of-memory issues when a large codebook size is employed.

## C CONVERGENCE BEHAVIOR OF SVQ AND TRADITIONAL VQ

SVQ and traditional VQ are based on sparse regression and clustering, respectively. Since SVQ is fully differentiable, their convergence behaviors can be considered analogous to Backpropagation (BP) and K-Nearest Neighbors (KNN), respectively. We prove that BP's optimization is smoother than KNN's due to continuous gradient descent.

**Backpropagation.** The decision function is directly related to the weights and activation functions, which tend to evolve smoothly under gradient descent.

$$f(x) = \sigma(W_n(\sigma(W_{n-1}(\ldots \sigma(W_1 x + b_1) \ldots) + b_{n-1})) + b_n), \tag{11}$$

with the gradient updates affecting $W_i$ and $b_i$:

$$W_i \leftarrow W_i - \eta \nabla_{W_i} J, \tag{12}$$

$$b_i \leftarrow b_i - \eta \nabla_{b_i} J. \tag{13}$$

The gradient $\nabla L(\mathbf{W})$ is typically smooth and continuous if the loss function $L$ and the activation functions are smooth. Hence, the weights update in a relatively smooth manner, and the path to the minimum of the loss function is traversed in small, continuous steps.

**K-Nearest Neighbors.** Being a non-parametric method, KNN directly relies on the training data to make its predictions. For a new input $\mathbf{x}$, the prediction $y$ is made based on the majority vote among its $k$ nearest neighbors:

$$y = \text{argmax} \sum_{i=1}^{k} \delta(y_i, y), \tag{14}$$

where $\delta$ is the Kronecker delta function.

The decision boundary of KNN is piecewise linear and can change abruptly with small changes in the input data. Consider the case where the input $\mathbf{x}$ moves slightly from one side of the decision boundary to the other:

$$\mathbf{x} \to \mathbf{x}' | \mathbf{x} \approx \mathbf{x}'. \tag{15}$$

In this case, the set of $k$ nearest neighbors might change abruptly, leading to a discontinuous jump in the prediction $y$.

We can conclude that the optimization process for backpropagation is smoother as compared to KNN due to the continuous nature of gradient descent, which updates the weights in small, incremental steps:

$$\mathbf{W}_{t+1} = \mathbf{W}_t - \eta \nabla L(\mathbf{W}_t). \tag{16}$$

In contrast, KNN's decision boundary can lead to discontinuous predictions with small perturbations in the input data, illustrating the non-smooth nature of the optimization process in KNN.

# D ADDITIONAL QUANTITATIVE RESULTS

## D.1 COMPUTATIONAL COST

Table 11 presents the computational costs of SVQ module and forecasting models. It shows that recurrent-based models have significantly higher FLOPS requirements, while non-recurrent models are more efficient. The proposed SVQ module is not only effective but also computationally cheap. Across all datasets, SVQ only slightly adds the number of parameters and FLOPS. The computational burden of SimVP+SVQ remains significantly smaller than recurrent-based models.

Table 11: Number of parameters and computing performances for all forecasting models.

| Model type | Dataset | Human3.6M | | KTH | | KittiCaltech | | WeatherBench-S | | TaxiBJ | |
|---|---|---|---|---|---|---|---|---|---|---|---|
| | Model | Params | FLOPS | Params | FLOPS | Params | FLOPS | Params | FLOPS | Params | FLOPS |
| Recurrent-based | ConvLSTM | 15.5M | 347.0G | 14.9M | 1368.0G | 15.0M | 595.0G | 14.98M | 136G | 14.98M | 20.74G |
| | E3D-LSTM | 60.9M | 542.0G | 53.5M | 217.0G | 54.9M | 1004G | 51.09M | 169G | 50.99M | 98.19G |
| | PredNet | 12.5M | 13.7G | 12.5M | 3.4G | 12.5M | 12.5M | - | - | 12.5M | 0.85G |
| | PhyDNet | 4.2M | 19.1G | 3.1M | 93.6G | 3.1M | 40.4G | 3.09M | 36.8G | 3.09M | 5.60G |
| | MAU | 20.2M | 105.0G | 20.1M | 399.0G | 24.3M | 172.0G | 5.46M | 39.6G | 4.41M | 6.02G |
| | MIM | 47.6M | 1051.0G | 39.8M | 1099.0G | 49.2M | 1858G | 37.75M | 109G | 37.86M | 64.10G |
| | PredRNN | 24.6M | 704.0G | 23.6M | 2800.0G | 23.7M | 1216G | 23.57M | 278G | 23.66M | 42.40G |
| | PredRNN++ | 39.3M | 1033.0G | 38.3M | 4162.0G | 38.5M | 1803G | 38.31M | 413G | 38.40M | 62.95G |
| | PredRNN.V2 | 24.6M | 708.0G | 23.6M | 2815.0G | 23.8M | 1223G | 23.59M | 279G | 23.67M | 42.63G |
| | DMVFN | - | - | 3.5M | 0.88G | 3.6M | 1.2G | - | - | 3.54M | 0.057G |
| Non-recurrent | TAU | 37.6M | 182.0G | 15.0M | 73.8G | 44.7M | 80.0G | 12.22M | 6.70G | 9.55M | 2.49G |
| | SimVP (w/o VQ) | 28.8M | 146.0G | 12.2M | 62.8G | 15.6M | 96.3G | 12.76M | 7.01G | 7.84M | 2.08G |
| | **SimVP+SVQ** | 30.7M | 178.0G | 13.3M | 110.0G | 16.8M | 156G | 14.37M | 16.8G | 9.45M | 3.72G |

## D.2 ROBUSTNESS TO ARTIFICIAL NOISE INJECTION

To clearly demonstrate the noise reduction effect of SVQ, we conduct a series of experiments by introducing controlled noise to training data in order to simulate perturbations. The fraction of the data to be perturbed is determined by $\eta$. The noise to be added is generated with uniform random values scaled to the range [-2,2]. Table 12 shows that the SVQ-equiped model experiences a much lower rise in MSE and MAE relative to the model without SVQ, regardless of the proportion of injected noise. For instance, when the proportion of injected noise is 10%, the MSE of the model without SVQ rises by 25.4%, whereas the model with SVQ shows a modest increase of just 3.2%. These results confirm the effect of SVQ on helping the forecasting model handle noise better through vector quantization.

Table 12: **Noise injection analysis:** The proportion of injected noise is indicated by $\eta$. We present MSE and MAE, and their percentage increase over baseline without artificial noise.

| Noise proportion | MSE | | MAE | |
|---|---|---|---|---|
| | w/o SVQ | w SVQ | w/o SVQ | w SVQ |
| $\eta$=0 | 1.105 | 1.018 | 0.6567 | 0.6109 |
| $\eta$=10% | 1.386(+25.4%) | 1.051(+3.2%) | 0.7702(+17.3%) | 0.6269(+2.6%) |
| $\eta$=20% | 1.554(+40.7%) | 1.196(+17.5%) | 0.8282(+26.1%) | 0.6710(+9.8%) |
| $\eta$=30% | 1.750(+58.4%) | 1.255(+23.2%) | 0.8821(+34.3%) | 0.6953(+13.8%) |
| $\eta$=40% | 2.081(+88.3%) | 1.568(+54.0%) | 1.0031(+52.7%) | 0.7973(+30.5%) |
| $\eta$=50% | 2.646(+139.4%) | 1.529(+50.2%) | 1.1339(+72.7%) | 0.7881(+29.0%) |

### D.3    Position of quantization module

To illustrate how the position of quantization module affects representation learning, we consider two designs as shown in Figure 8. In our main experiments, we adopt the first design where quantization is performed before the translator. In Section 5.4, we also investigate an alternative design where quantization is performed after the translator. Two designs only differ in the placement order of quantization module and translator module, while the other settings are kept the same.

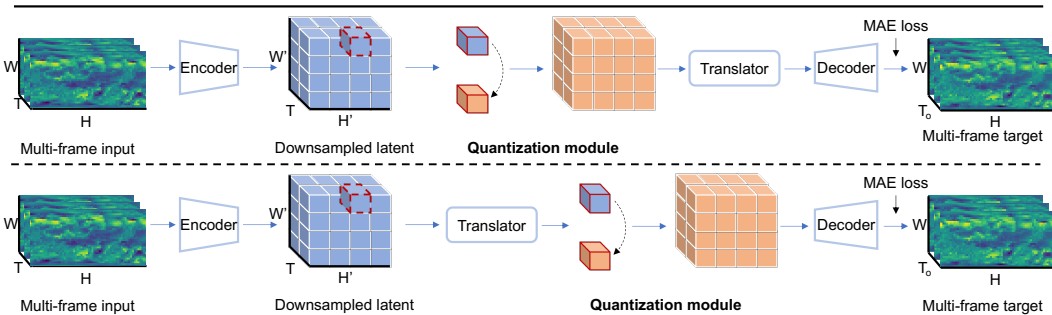

Figure 8: Comparison of two quantization designs with different positions. Top: Quantization before translator. Bottom: Quantization after translator.

### D.4    Comprehensive comparison of VQ methods using the same codebook size

To make a fair comparison, we extend Table 4 by setting the codebook size to the same value (1024) for compared VQ methods. They are comprehensively evaluated from different aspects including downstream performance (prediction MSE), codebook usage (perplexity), and computational complexity (FLOPS, inference FPS, and training time per epoch). The quantitative results are shown in Table 13. The convergence performance is shown in Figure 9, where SVQ quickly converges to the lowest prediction error and satisfactory utilization of the codebook. Residual VQ with stochastic sampling has the highest codebook usage. However, its prediction MSE is worse than residual VQ without stochastic sampling. This demonstrates that forcibly improving codebook usage does not guarantee better downstream performance. SVQ generally outperforms the other VQ methods in computational efficiency, due to the simplified approximation described in Section 3.1.

Table 13: Quantitative comparison of VQ methods with the same codebook size (1024) on WeatherBench-S temperature dataset.

| Vector quantization method | Prediction MSE↓ | Perplexity↑ | FLOPS↓ | Inference FPS↑ | Training time per epoch(min)↓ |
|---|---|---|---|---|---|
| VQ | 1.8544 | 51.95 | 7.207G | 21.1 | 7.11 |
| Residual VQ | 1.2131 | 142.47 | 8.616G | 7.7 | 13.25 |
| Residual VQ (Stochastic) | 1.8882 | 817.91 | 8.616G | 8.1 | 17.27 |
| Grouped Residual VQ | 1.1737 | 132.57 | 8.616G | 4.8 | 19.98 |
| Multi-headed VQ | 1.2113 | 16.36 | 8.717G | 6.2 | 13.15 |
| SVQ (Frozen) | 1.0393 | 335.72($\theta$=3)/438.39($\theta$=2) | 8.037G | 24.6 | 7.27 |
| SVQ (Learnable) | 1.0403 | 246.44($\theta$=3)/331.41($\theta$=2) | 8.037G | 24.9 | 7.30 |

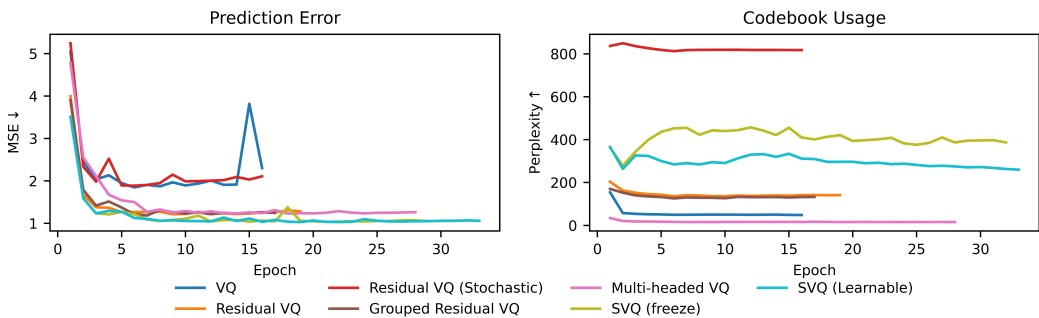

Figure 9: Prediction error and codebook usage of different VQ methods during the training process. All methods adopt the same codebook size (1024) and are early stopped with a patience of 10, trained on WeatherBench-S temperature dataset. For simplicity, the perplexity of SVQ is averaged on different $\theta$.

## D.5 ABLATION OF FROZEN MODULE

The SVQ module consists of a two-layer MLP and a large codebook. The MLP can be seen as a projection from the input vector to the regression weights. In Section 5.5, we have already examined the impact of freezing the codebook on forecasting performance. To further investigate the effect of freezing the two-layer MLP, we conducted an ablation study in this section. The results, presented in Table 14, show that freezing the codebook only has a slight impact on forecasting performance, while freezing the MLP significantly impairs the performance. The MLP is essential for sparse regression and must be learned from the data, as it generates the weights needed to combine codes from the codebook.

Table 14: Ablation of frozen modules on WeatherBench-S temperature dataset.

| Metric | Frozen module | | | |
|---|---|---|---|---|
| | None (All learnable) | Codebook | Two-layer MLP projection | Both |
| MSE | **1.018** | 1.023 | 1.060 | 1.093 |
| MAE | **0.6109** | 0.6131 | 0.6194 | 0.6387 |

## D.6 EXPERIMENT STATISTICAL SIGNIFICANCE

To get more robust experimental results and evaluate the statistical significance, we rerun SimVP and SimVP+SVQ models five times under identical conditions. The results are presented without standard deviations in the main text due to space constraints. The results with standard deviations on the WeatherBench-S temperature, TaxiBJ, and WeatherBench-M datasets are reported in Tables 15, 16, and 17, respectively. The standard deviations of the SimVP+SVQ model are generally smaller than or comparable to those of the SimVP model.

Table 15: Statistical significance of models on the WeatherBench-S temperature dataset.

| Model | MSE↓ | MAE↓ |
|---|---|---|
| SimVP (w/o VQ) | 1.105±0.043 | 0.6567±0.0185 |
| SimVP+SVQ (Frozen) | 1.023±0.007 | 0.6131±0.0050 |
| SimVP+SVQ (Learnable) | 1.018±0.009 | 0.6109±0.0064 |

Table 16: Statistical significance of models on the TaxiBJ dataset.

| Model | MSE↓ | MAE↓ | SSIM↑ | PSNR↑ |
|---|---|---|---|---|
| SimVP (w/o VQ) | 0.3246±0.0173 | 15.03±0.26 | 0.9844±0.0008 | 39.71±0.11 |
| SimVP+SVQ (Frozen) | 0.3171±0.0056 | 14.68±0.03 | 0.9848±0.0001 | 39.83±0.01 |
| SimVP+SVQ (Learnable) | 0.3191±0.0020 | 14.64±0.02 | 0.9849±0.0002 | 39.86±0.02 |

Table 17: Statistical significance of models on the WeatherBench-M dataset.

| Variable | Temperature | | Humidity | | Wind Component | | Total Cloud Cover | |
|---|---|---|---|---|---|---|---|---|
| Model | MSE↓ | MAE↓ | MSE↓ | MAE↓ | MSE↓ | MAE↓ | MSE↓ | MAE↓ |
| SimVP (w/o VQ) | 4.833±0.031 | 1.5246±0.0077 | 340.06±0.44 | 12.738±0.030 | 24.535±0.076 | 3.4882±0.0086 | 25.332±0.052 | 3.4509±0.0052 |
| SimVP+SVQ (Frozen) | 4.427±0.017 | 1.4160±0.0051 | 360.15±0.55 | 12.445±0.024 | 23.915±0.098 | 3.4078±0.0033 | 24.968±0.126 | 3.4117±0.0032 |
| SimVP+SVQ (Learnable) | 4.433±0.021 | 1.4164±0.0054 | 360.53±0.81 | 12.449±0.013 | 23.908±0.062 | 3.4060±0.0029 | 24.983±0.115 | 3.4095±0.0038 |

## D.7 BENCHMARK ON TAXIBJ DATASET

Table 18: Performance comparison for SVQ and baseline models on TaxiBJ.

| Model | MSE↓ | MAE↓ | SSIM↑ | PSNR↑ |
|---|---|---|---|---|
| ConvLSTMSHI et al. (2015) | 0.3358 | 15.32 | 0.9836 | 39.45 |
| E3D-LSTMWang et al. (2019a) | 0.3427 | 14.98 | 0.9842 | 39.64 |
| PhyDNetGuen & Thome (2020) | 0.3622 | 15.53 | 0.9828 | 39.46 |
| PredNetLotter et al. (2017) | 0.3516 | 15.91 | 0.9828 | 39.29 |
| PredRNNWang et al. (2017) | 0.3194 | 15.31 | 0.9838 | 39.51 |
| MIMWang et al. (2019b) | 0.3110 | 14.96 | 0.9847 | 39.65 |
| MAUChang et al. (2021) | 0.3268 | 15.26 | 0.9834 | 39.52 |
| DMVFNHu et al. (2023) | 3.3954 | 45.52 | 0.8321 | 31.14 |
| PredRNN++Wang et al. (2018) | 0.3348 | 15.37 | 0.9834 | 39.47 |
| PredRNN.V2Wang et al. (2022) | 0.3834 | 15.55 | 0.9826 | 39.49 |
| TAUTan et al. (2023a) | **0.3108** | 14.93 | 0.9848 | 39.74 |
| SimVP (w/o VQ)Tan et al. (2022) | 0.3246 | 15.03 | 0.9844 | 39.71 |
| **SimVP+SVQ (Frozen)** | 0.3171 | 14.68 | 0.9848 | 39.83 |
| **SimVP+SVQ (Learnable)** | 0.3191 | **14.64** | **0.9849** | **39.86** |
| **Improvement** | ↑**1.7%** | ↑**2.6%** | ↑**0.1%** | ↑**0.4%** |

## D.8 BENCHMARK ON WEATHERBENCH-HMV DATASET

To evaluate the performance of SVQ on high-dimensional data, we conducted additional experiments utilizing the WeatherBench dataset Rasp et al. (2020) in a High-dimensional Multi-Variable (HMV) setting. This dataset, related to real-world weather forecasting, consists of various meteorological variables that contribute to a total of 110 channels. These include temperature at 2 m height above surface (t2m), wind in x/longitude-direction at 10 m height (u10), accumulated hourly incident solar radiation (tisr), fractional cloud cover (tcc), hourly precipitation (tp), potential vorticity (pv), etc. Notably, several variables are structured across multiple vertical layers. For instance, pv_50 denotes the potential vorticity at 50 hPa. The WeatherBench-HMV dataset is similar to WeatherBench-M but includes significantly more channels (increasing from 4 to 110). It is designed for multi-variable, six-hour interval forecasting, trained on data from 1980-2015, validated on data from 2016, and tested on data from 2017-2018. Owing to constraints on page length, we divided the performance metrics of SimVP and SVQ across these 110 channels into two separate tables, as detailed in Tables 19 and 20. We observed that SVQ achieved an average enhancement of 11.7% on the initial 55 channels, while the improvement on the subsequent 55 channels was 6.1%. Consequently, the cumulative average improvement across all 110 channels was 8.9%. These findings underscore that, despite the complexities introduced by high-dimensional datasets, SVQ effectively adapts to and enhances the predictive capabilities of the backbone model.

Table 19: MAE comparison for SimVP and SVQ on WeatherBench-HMV (The first 55 channels).

| Channel | SimVP (w/o VQ) | SimVP+SVQ (Frozen) | SimVP+SVQ (Learnable) | Improvement |
|---------|----------------|--------------------|-----------------------|-------------|
| u10 | 2.1557 | 2.0849 | 2.0757 | 3.7% |
| v10 | 2.1613 | 2.1101 | 2.1117 | 2.4% |
| t2m | 2.5603 | 2.4089 | 2.2799 | 11.0% |
| tisr | 1.13E+05 | 32640 | 30756 | 72.7% |
| tcc | 0.22906 | 0.22028 | 0.22316 | 3.8% |
| tp | 0.000147 | 9.96E-05 | 9.96E-05 | 32.1% |
| z_50 | 371.19 | 268.81 | 269.45 | 27.6% |
| z_100 | 333.31 | 243.12 | 245.07 | 27.1% |
| z_150 | 339.15 | 265.88 | 267.14 | 21.6% |
| z_200 | 376.12 | 303.11 | 302.84 | 19.5% |
| z_250 | 404.46 | 340.07 | 338.73 | 16.3% |
| z_300 | 404.8 | 347.96 | 347.64 | 14.1% |
| z_400 | 357.33 | 314.15 | 313.48 | 12.3% |
| z_500 | 309.93 | 272.9 | 270.52 | 12.7% |
| z_600 | 276.35 | 241.09 | 237.15 | 14.2% |
| z_700 | 252.58 | 218.67 | 215.86 | 14.5% |
| z_850 | 229.2 | 203.88 | 203.05 | 11.4% |
| z_925 | 229 | 208.52 | 206.46 | 9.8% |
| z_1000 | 242.2 | 220.18 | 219.18 | 9.5% |
| pv_50 | 3.97E-06 | 2.93E-06 | 2.92E-06 | 26.3% |
| pv_100 | 1.44E-06 | 1.32E-06 | 1.33E-06 | 7.8% |
| pv_150 | 9.72E-07 | 8.42E-07 | 8.44E-07 | 13.4% |
| pv_200 | 1.00E-06 | 8.64E-07 | 8.50E-07 | 15.0% |
| pv_250 | 1.05E-06 | 9.77E-07 | 9.73E-07 | 7.7% |
| pv_300 | 8.51E-07 | 7.88E-07 | 8.00E-07 | 7.4% |
| pv_400 | 3.61E-07 | 3.42E-07 | 3.41E-07 | 5.6% |
| pv_500 | 2.35E-07 | 2.28E-07 | 2.28E-07 | 3.1% |
| pv_600 | 3.89E-07 | 3.31E-07 | 3.32E-07 | 14.8% |
| pv_700 | 6.97E-07 | 5.30E-07 | 5.27E-07 | 24.4% |
| pv_850 | 8.81E-07 | 7.70E-07 | 7.62E-07 | 13.6% |
| pv_925 | 1.00E-06 | 9.66E-07 | 9.58E-07 | 4.6% |
| pv_1000 | 1.40E-06 | 1.35E-06 | 1.34E-06 | 4.5% |
| r_50 | 1.9046 | 1.2558 | 1.391 | 34.1% |
| r_100 | 7.6728 | 6.0793 | 6.3087 | 20.8% |
| r_150 | 9.6361 | 8.4854 | 8.7663 | 11.9% |
| r_200 | 14.653 | 13.175 | 14.435 | 10.1% |
| r_250 | 19.433 | 18.728 | 18.964 | 3.6% |
| r_300 | 20.066 | 19.791 | 19.851 | 1.4% |
| r_400 | 19.78 | 19.375 | 19.356 | 2.1% |
| r_500 | 19.155 | 18.876 | 18.825 | 1.7% |
| r_600 | 18.208 | 17.981 | 17.938 | 1.5% |
| r_700 | 17.216 | 17.041 | 17.055 | 1.0% |
| r_850 | 14.586 | 15.379 | 14.691 | -0.7% |
| r_925 | 10.086 | 9.613 | 9.6737 | 4.7% |
| r_1000 | 7.5008 | 8.0927 | 7.2911 | 2.8% |
| q_50 | 9.03E-08 | 7.40E-08 | 7.55E-08 | 18.1% |
| q_100 | 1.97E-07 | 1.87E-07 | 1.87E-07 | 5.3% |
| q_150 | 1.10E-06 | 1.10E-06 | 1.08E-06 | 2.1% |
| q_200 | 6.68E-06 | 6.48E-06 | 6.39E-06 | 4.3% |
| q_250 | 2.38E-05 | 2.24E-05 | 2.26E-05 | 5.7% |
| q_300 | 5.61E-05 | 5.30E-05 | 5.40E-05 | 5.5% |
| q_400 | 0.000181 | 0.000174 | 0.000177 | 3.9% |
| q_500 | 0.000375 | 0.000364 | 0.000362 | 3.4% |
| q_600 | 0.000592 | 0.000557 | 0.000553 | 6.6% |
| q_700 | 0.000826 | 0.000775 | 0.000784 | 6.2% |
| **Average improvement** | | | | **11.7%** |

Table 20: MAE comparison for SimVP and SVQ on WeatherBench-HMV (The last 55 channels).

| Channel | SimVP (w/o VQ) | SimVP+SVQ (Frozen) | SimVP+SVQ (Learnable) | Improvement |
|---|---|---|---|---|
| q_850 | 0.001054 | 0.000989 | 0.000994 | 6.1% |
| q_925 | 0.0009 | 0.000757 | 0.00075 | 16.7% |
| q_1000 | 0.001024 | 0.000776 | 0.000767 | 25.1% |
| t_50 | 1.7426 | 1.2564 | 1.3594 | 27.9% |
| t_100 | 1.5365 | 1.2852 | 1.3544 | 16.4% |
| t_150 | 1.5825 | 1.3185 | 1.3377 | 16.7% |
| t_200 | 1.7535 | 1.6099 | 1.6243 | 8.2% |
| t_250 | 1.7895 | 1.5499 | 1.5571 | 13.4% |
| t_300 | 1.5555 | 1.3903 | 1.3658 | 12.2% |
| t_400 | 1.6055 | 1.4029 | 1.3737 | 14.4% |
| t_500 | 1.6277 | 1.4652 | 1.4493 | 11.0% |
| t_600 | 1.693 | 1.5669 | 1.5636 | 7.6% |
| t_700 | 1.8961 | 1.7623 | 1.7412 | 8.2% |
| t_850 | 2.1494 | 2.0246 | 1.8345 | 14.6% |
| t_925 | 2.2241 | 1.8933 | 1.8747 | 15.7% |
| t_1000 | 2.2335 | 1.891 | 1.834 | 17.9% |
| u_50 | 2.9882 | 2.5862 | 2.5981 | 13.5% |
| u_100 | 3.4492 | 3.3665 | 3.3759 | 2.4% |
| u_150 | 3.9731 | 3.7005 | 3.7262 | 6.9% |
| u_200 | 4.8379 | 4.597 | 4.5959 | 5.0% |
| u_250 | 5.725 | 5.5288 | 5.5195 | 3.6% |
| u_300 | 5.9494 | 5.7551 | 5.7492 | 3.4% |
| u_400 | 5.1291 | 4.9461 | 4.9373 | 3.7% |
| u_500 | 4.2227 | 4.0826 | 4.0643 | 3.8% |
| u_600 | 3.7165 | 3.6228 | 3.5999 | 3.1% |
| u_700 | 3.4591 | 3.3916 | 3.3759 | 2.4% |
| u_850 | 3.0929 | 3.0253 | 3.0211 | 2.3% |
| u_925 | 2.9711 | 2.8879 | 2.8847 | 2.9% |
| u_1000 | 2.3824 | 2.298 | 2.2888 | 3.9% |
| v_50 | 2.4844 | 2.3521 | 2.346 | 5.6% |
| v_100 | 3.0661 | 2.9852 | 2.9837 | 2.7% |
| v_150 | 3.8109 | 3.6697 | 3.6702 | 3.7% |
| v_200 | 4.7779 | 4.6044 | 4.5961 | 3.8% |
| v_250 | 5.6621 | 5.5064 | 5.4953 | 2.9% |
| v_300 | 5.8984 | 5.7684 | 5.7591 | 2.4% |
| v_400 | 5.0538 | 4.9421 | 4.9351 | 2.3% |
| v_500 | 4.1298 | 4.0279 | 4.0228 | 2.6% |
| v_600 | 3.612 | 3.5264 | 3.5224 | 2.5% |
| v_700 | 3.3112 | 3.2528 | 3.2485 | 1.9% |
| v_850 | 3.0267 | 2.9877 | 2.9857 | 1.4% |
| v_925 | 2.9499 | 2.8942 | 2.8931 | 1.9% |
| v_1000 | 2.3969 | 2.3354 | 2.3381 | 2.6% |
| vo_50 | 8.64E-06 | 8.50E-06 | 8.50E-06 | 1.6% |
| vo_100 | 1.16E-05 | 1.15E-05 | 1.15E-05 | 0.5% |
| vo_150 | 1.59E-05 | 1.58E-05 | 1.58E-05 | 0.9% |
| vo_200 | 2.26E-05 | 2.24E-05 | 2.24E-05 | 0.8% |
| vo_250 | 3.24E-05 | 3.22E-05 | 3.22E-05 | 0.6% |
| vo_300 | 3.85E-05 | 3.82E-05 | 3.82E-05 | 0.9% |
| vo_400 | 3.61E-05 | 3.58E-05 | 3.57E-05 | 1.0% |
| vo_500 | 2.99E-05 | 2.96E-05 | 2.96E-05 | 0.8% |
| vo_600 | 2.69E-05 | 2.67E-05 | 2.68E-05 | 0.8% |
| vo_700 | 2.68E-05 | 2.65E-05 | 2.65E-05 | 1.1% |
| vo_850 | 2.76E-05 | 2.76E-05 | 2.76E-05 | 0.2% |
| vo_925 | 2.64E-05 | 2.63E-05 | 2.63E-05 | 0.2% |
| vo_1000 | 2.08E-05 | 2.08E-05 | 2.08E-05 | 0.1% |
| **Average improvement** | | | | **6.1**% |

## E EFFECT OF SVQ ON LATENT REPRESENTATION

We investigated the impact of SVQ on the sparsity of regression weights in Figure 7. To further investigate its effect on the latent representation, we compared the distribution of batch tensors before and after applying SVQ. The tensors were transformed into normalized vectors, and their density distributions were estimated. As depicted in Figure 10, the representation after SVQ demonstrates a more compact distribution, indicating improved robustness to noise. These results further prove that SVQ can enhance forecasting performance by effectively handling noise in the data.

Figures 11, 12, 14, 15, and 13 present the comparison of latent feature maps before and after applying SVQ. These figures illustrate that the difference between foreground and background in the feature maps increases after SVQ. For example, in the KittiCaltech dataset, a clear distinction is observed between road conditions and sky (Figure 11). Similarly, in the WeatherBench-S temperature dataset, distinctive regions are identified between high and low latitudes (Figure 12). These findings suggest that SVQ helps in enhancing the discriminative power of the latent representations, which in turn contributes to improved downstream forecasting performance.

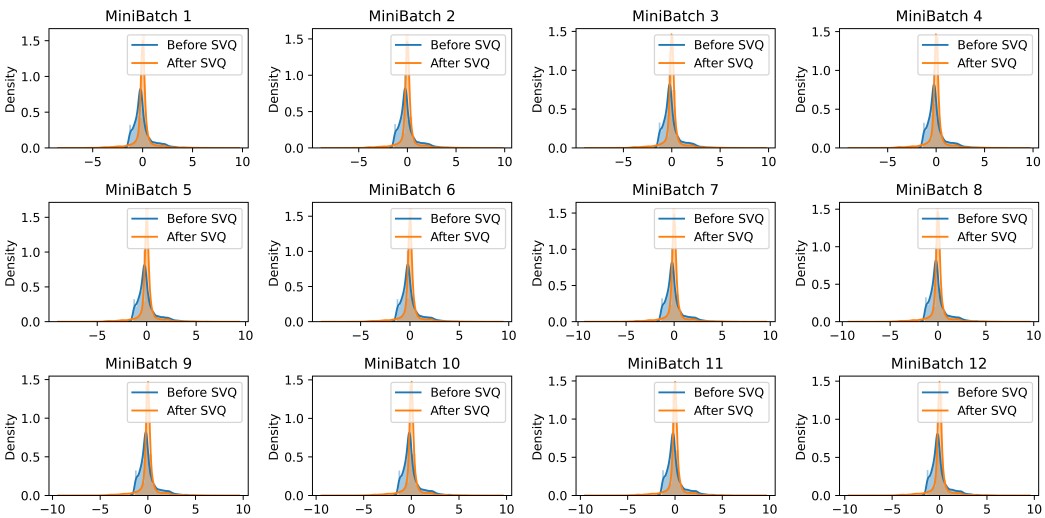

Figure 10: Distribution of latent vector on WeatherBench-S temperature dataset.

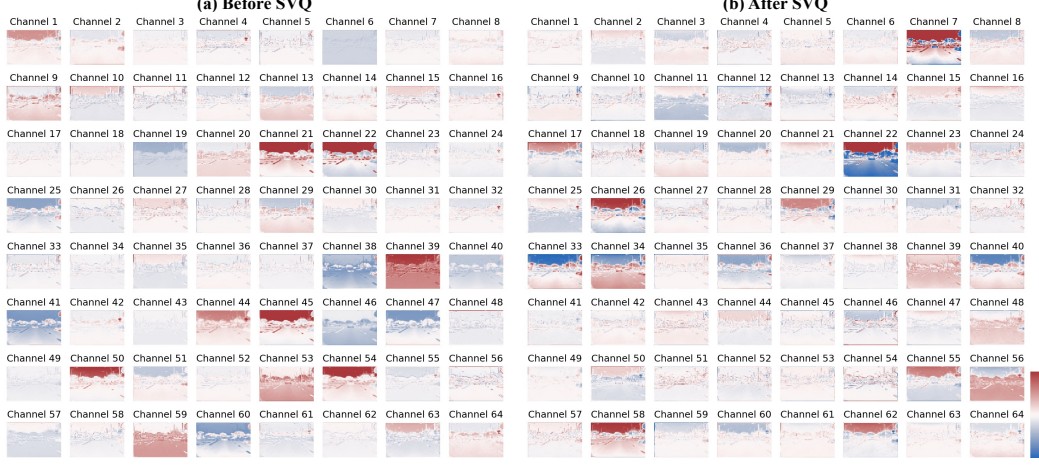

Figure 11: Latent feature map on the KittiCaltech dataset: (a) feature map before SVQ, and (b) feature map after SVQ.

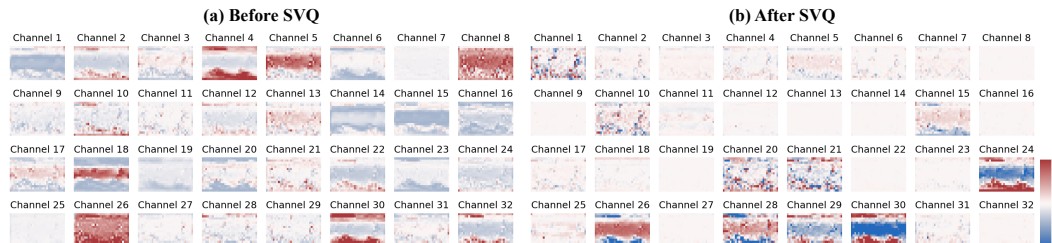

Figure 12: Latent feature map on the WeatherBench-S temperature dataset: (a) feature map before SVQ, and (b) feature map after SVQ.

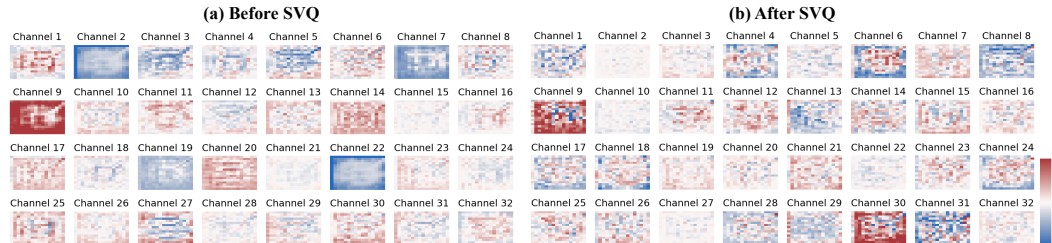

Figure 13: Latent feature map on the TaxiBJ dataset: (a) feature map before SVQ, and (b) feature map after SVQ.

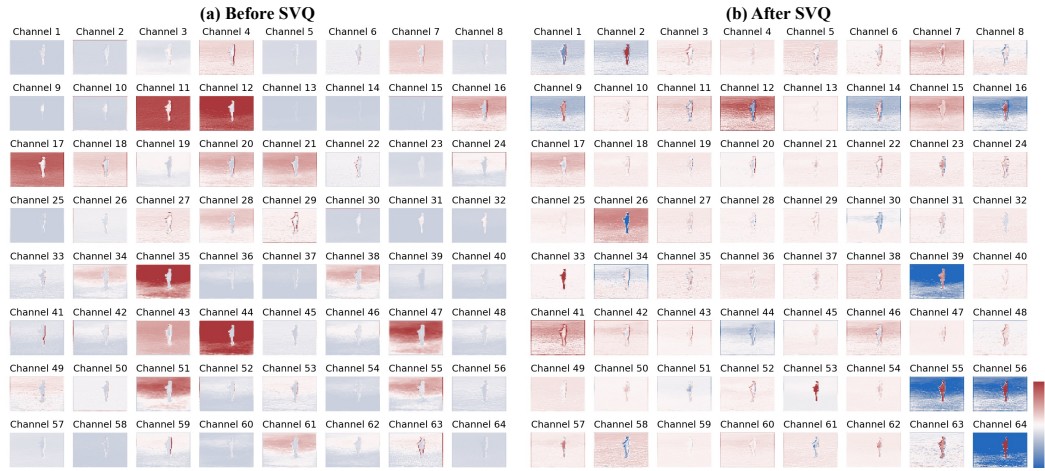

Figure 14: Latent feature map on the KTH dataset: (a) feature map before SVQ, and (b) feature map after SVQ.

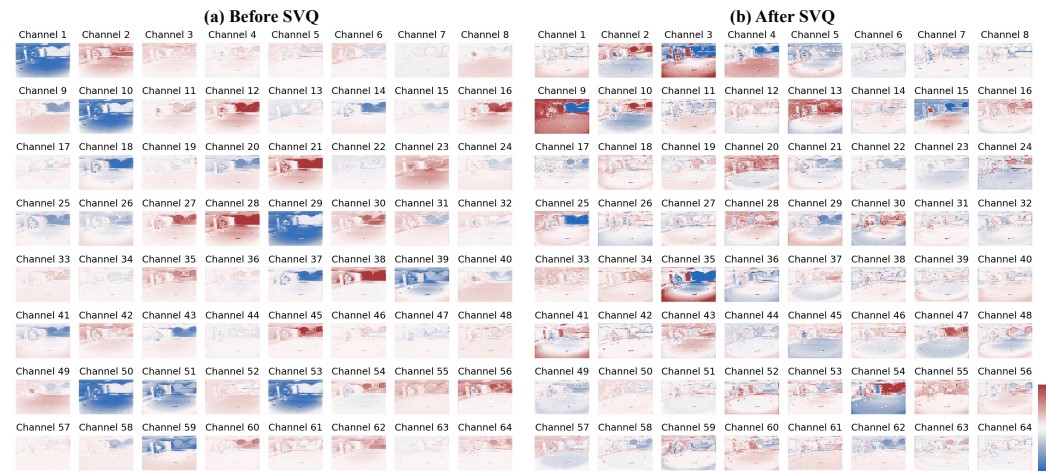

Figure 15: Latent feature map on the Human3.6M dataset: (a) feature map before SVQ, and (b) feature map after SVQ.

## F  ADDITIONAL QUALITATIVE RESULTS

### F.1  FORECASTING ERRORS ON WEATHERBENCH AND TAXIBJ DATASETS

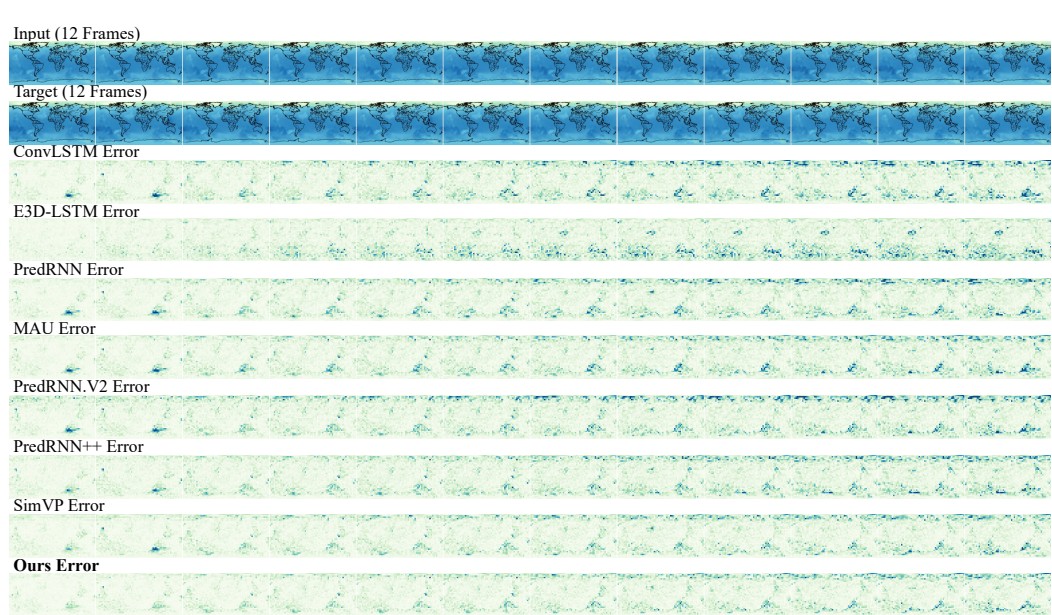

Figure 16: The qualitative forecasting errors on WeatherBench-S temperature dataset.

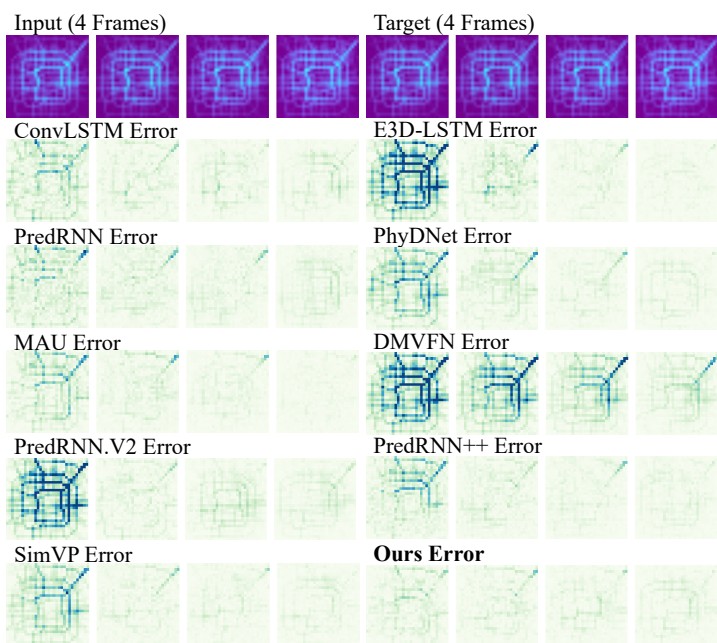

Figure 17: The qualitative forecasting errors on TaxiBJ dataset.

## F.2 ADDITIONAL FORECASTING SAMPLES

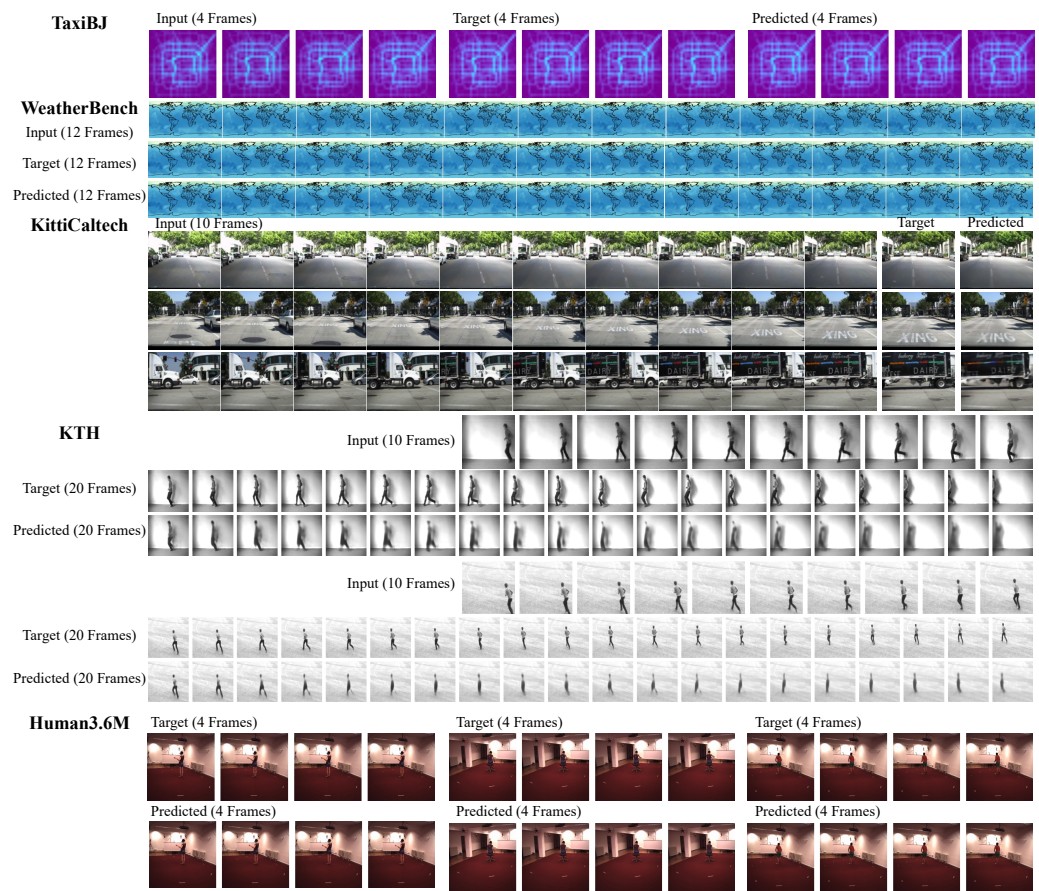

Figure 18: Forecasting samples of SimVP+SVQ model on the test set of TaxiBJ (32×32), Weather-Bench (64×32), KittiCaltech (160×128), KTH (128×128), and Human3.6M (256×256). Zoom in for details. Our model produces accurate predicted frames for different tasks.

