# OpenReview forum: "Does Vector Quantization Fail in Spatio-Temporal Forecasting? Exploring a Differentiable Sparse Soft-Vector Quantization Approach"
_ICLR.cc/2025/Conference — Submitted to ICLR 2025_

### Official Review · Reviewer_1Zsw · 2024-10-30

**Soundness:** 3
**Presentation:** 4
**Contribution:** 4
**Rating:** 6
**Confidence:** 5

**Summary:**

This paper addresses the limitations of traditional Vector Quantization (VQ) and demonstrates impressive performance in spatio-temporal forecasting. SVQ uses a two-layer MLP to approximate sparse regression, reducing computational complexity while maintaining the flexibility to map each input vector to multiple codebook vectors. This soft quantization captures the complex dynamics of spatio-temporal data, preserving essential information while minimizing noise. The experiments confirm that SVQ is an efficient and expressive quantization mechanism applicable to various forecasting tasks. The visualizations provide valuable insights into the behavior and advantages of SVQ.

**Strengths:**

1. Innovative Approach: The paper effectively combines sparse regression with differentiable quantization, addressing the non-differentiability and limited representational power of traditional VQ. Using MLP to approximate sparse regression allows the model to capture complex patterns efficiently.
2. Simplicity and Effectiveness: The proposed method is intuitive and easy to implement, with straightforward derivations and motivations. It demonstrates significant improvements across multiple tasks and models.
3. Comprehensive Experiments: The paper provides detailed evaluations of the proposed quantization mechanism, including ablation studies and supplementary materials that address key questions. The well-designed visualizations offer excellent insights into the behavior and strengths of SVQ.

**Weaknesses:**

1. Visual Layout: Perhaps due to space constraints, the layout of the figures and tables could be more aesthetically pleasing.

**Questions:**

1. The paper emphasizes the advantages of the proposed method in spatio-temporal forecasting, but VQ is also widely used in generative tasks (e.g., VQ-VAE). Could this method be applied to such tasks?

---

> ### Author Response · Authors · 2024-11-23
> **Response to Reviewer 1Zsw**
>
> Dear Reviewer 1ZsW,
>
> We sincerely thank your thoughtful insights and recognition of our efforts in exploring VQ's deterioration effect in spatio-temporal forecasting and for proposing SVQ. Your support and endorsement of our work truly inspire us to continue pursuing this research direction.
>
> > **[W1]** Visual Layout: Perhaps due to space constraints, the layout of the figures and tables could be more aesthetically pleasing.
>
> We acknowledge that the original arrangement might have felt constrained due to space limitations. In response, we have carefully revised the layout to make it more visually appealing. The revised version of our paper is available for download.
>
> > **[Q1]** The paper emphasizes the advantages of the proposed method in spatio-temporal forecasting, but VQ is also widely used in generative tasks (e.g., VQ-VAE). Could this method be applied to such tasks?
>
> We anticipate that our proposed method will likely improve image generation due to its broad applicability in addressing non-differentiability and representation issues that are generally unrelated to the design of base models. We are eager to test our method in domains such as image generation, multimodal LLMs, and other application scenarios where existing VQ blocks provide a natural substitution. However, many of these tasks are extremely GPU-intensive, and we currently lack the resources to pursue them. Spatio-temporal forecasting, on the other hand, is an application that does not require tens or hundreds of GPUs, making it a viable option for exploration.
>
> The OpenSTL [1] benchmark encompasses three primary categories: weather forecasting, video forecasting, and traffic forecasting. In the realm of video forecasting, there is a strong focus on evaluating the quality of output images, moving beyond traditional metrics like MSE and MAE, which have been the standard measures in both weather and traffic forecasting. Our work demonstrates improvement across all three tasks in terms of MSE/MAE metrics as well as image quality metrics. For instance, using the LPIPS metric, SVQ demonstrates enhancements of 26.2\%, 3.6\%, and 22.2\% on three video forecasting datasets: Human3.6M, KittiCaltech, and KTH. We believe that our proposed method holds great potential for generative tasks.
>
> [1] Tan C, Li S, Gao Z, et al. Openstl: A comprehensive benchmark of spatio-temporal predictive learning[J]. Advances in Neural Information Processing Systems, 2023, 36: 69819-69831.

---

### Official Review · Reviewer_AkZ2 · 2024-11-01

**Soundness:** 2
**Presentation:** 3
**Contribution:** 2
**Rating:** 5
**Confidence:** 4

**Summary:**

The paper introduces a novel Differentiable Sparse Soft-Vector Quantization (SVQ) method, which integrates sparse regression with differentiability to tackle optimization challenges in vector quantization. This approach aims to enhance representation capacity in spatio-temporal forecasting tasks, marking a significant advancement in the field. While the SVQ method presents an innovative approach to vector quantization, the paper would benefit from clearer connections between theory and practice, updated baselines, deeper integration insights, and improved mathematical clarity to strengthen its contributions to the field.
Soundness:2

**Strengths:**

The proposed Differentiable Sparse Soft-Vector Quantization (SVQ) method represents a novel advancement in vector quantization techniques.

**Weaknesses:**

The method does not adequately demonstrate how the theoretical advantages of sparse regression are translated into tangible improvements in quantization performance. While the authors discuss optimization strategies, they fail to provide a clear connection between these theoretical claims and the practical outcomes. A more comprehensive explanation of how these optimization techniques directly enhance quantization would bolster the credibility of their approach.
	In the experimental section, I noticed that the baseline models employed are relatively outdated. Given the recent advancements in spatio-temporal forecasting, particularly the emergence of various diffusion-based methods that have demonstrated significant improvements in predictive performance, it would be beneficial for the authors to consider incorporating these state-of-the-art models as baselines. This would provide a more comprehensive evaluation of the proposed method's effectiveness and advantages.
	The explanation of the quantization module's implementation lacks depth regarding its integration with the overall spatio-temporal forecasting model. While the authors outline the architecture and components involved, they do not provide sufficient details on how the quantization process interacts with other model elements or influences the final forecasting results. A more thorough exploration of these interactions would enhance the clarity and applicability of their proposed method.
	In Section 4, the mathematical proof lacks clarity in the notation used, which may hinder readers' understanding. For example, what’s the meaning of g' after Eq. (8)? Additionally, the proof does not establish a strong connection to the problem being addressed. I recommend revising this section to improve the clarity of the symbols and to explicitly link the proof to the main objectives of the paper.

**Questions:**

See weakness

---

> ### Author Response · Authors · 2024-11-23
> **Response to Reviewer AkZ2**
>
> Dear Reviewer AkZ2,
>
> We sincerely appreciate the time and effort you have taken to provide such an insightful review. Your constructive feedback is invaluable to us. Below, we would like to address each point raised.
>
> > **[Q1]** The method does not adequately demonstrate how the theoretical advantages of sparse regression are translated into tangible improvements in quantization performance. While the authors discuss optimization strategies, they fail to provide a clear connection between these theoretical claims and the practical outcomes. A more comprehensive explanation of how these optimization techniques directly enhance quantization would bolster the credibility of their approach. In Section 4, the mathematical proof lacks clarity in the notation used, which may hinder readers' understanding. For example, what’s the meaning of g' after Eq. (8)? Additionally, the proof does not establish a strong connection to the problem being addressed. I recommend revising this section to improve the clarity of the symbols and to explicitly link the proof to the main objectives of the paper.
>
> Our theoretical analysis shows how the proposed VQ mimics the sparse regression model (i.e. ReLU and two layers in MLP correspond exactly to thresholding and quadratic dependence on fixed dictionary in sparse regression), and thus enjoys the advantage of sparse regression (i.e. error depends on $\log d$ not $d$), making it surpass traditional cluster-based VQ methods. Another key insight is that a frozen codebook can be nearly as effective as a trainable one, as long as it is sufficiently large. Although it may appear counterintuitive, our proof is based on a simple fact that any vector in a simplex can be well approximated by a sparse combination of vectors from a frozen and large dictionary. Thus, when using high-dimensional vectors, a sparse representation can be more expressive than a dense one in lower-dimensional space. Furthermore, for high-dimensional vectors, a basis can often be randomly sampled due to the high probability of orthogonality, eliminating the need for it to be learned. I hope this helps clarify the theoretical analysis, as these concepts are crucial for addressing the underlying 'why' questions.
>
> What we tried to show in the paper is that the first iteration of the sparse regression algorithm can be very implemented by a two layer MLP, where the nonlinear ReLU function in MLP corresponds to the thresholding used in the sparse regression algorithm, and the two layers corresponds to the quadratic dependence on dictionary matrix $Z$ in the algorithm. All the mathematical equivalence is illustrated in Eq. (2) and (3). Finally, it is not difficult to observe that we can simulate running the sparse regression algorithm with more than one step by introducing more layers (of course, have to be even number of layers) in MLP.
>
> g is an s-sparse unit vector, we use it to show the covering number. g' after Eq. (8) is a sample of the vectors from the set that densely covers the unit ball of  s-sparse vectors such that any s-sparse vector g can be closely approximated within this covering set. This is relevant to the discussion of vector covering numbers, which aim to cover a space with vectors such that every point in the space is close to at least one of these vectors. To bound $T(\mathcal{B}, \delta)$ for the clustering method, we use the covering number for a unit ball $\mathcal{B}$, requiring at least $1/\delta^d$ code vectors to approximate any vector within error $\delta$.

---

> ### Author Response · Authors · 2024-11-23
> **Continue Response to Reviewer AkZ2**
>
> > **[Q2]** In the experimental section, I noticed that the baseline models employed are relatively outdated. Given the recent advancements in spatio-temporal forecasting, particularly the emergence of various diffusion-based methods that have demonstrated significant improvements in predictive performance, it would be beneficial for the authors to consider incorporating these state-of-the-art models as baselines.
>
> Thank you for your suggestion. We are currently adhering to the OpenSTL benchmark [1], the ongoing benchmark for spatio-temporal forecasting. In light of this, we would like to include three additional baselines from the past two years: MMVP (ICCV 2023) [2], SwinLSTM (ICCV 2023) [3], and WaST (AAAI 2024) [4]. MMVP and WaST are evaluated on WeatherBench-S dataset. Due to SwinLSTM only supports H = W for image input, we evaluate it on TaxiBJ dataset. The results are presented in the table below. SVQ consistently enhances the performance of these baselines. The results are presented in the table below. SVQ consistently enhances the performance of these baselines. Furthermore, given the general nature of our proposed method, which addresses non-differentiability and representation issues typically orthogonal to base model design, we believe it is likely to enhance future methodologies.
>
> **Table: Additional baselines on WeatherBench-S dataset**
>
> | Model      | Temp (MSE↓) | Temp (MAE↓) | Humid (MSE↓) | Humid (MAE↓) | Wind (MSE↓) | Wind (MAE↓) | Cloud (MSE↓) | Cloud (MAE↓) |
> |------------|-------------|-------------|--------------|--------------|-------------|-------------|--------------|--------------|
> | WaST       | 1.239       | 0.686       | 33.349       | 3.881        | 1.603       | 0.838       | 0.048        | 0.150        |
> | WaST+SVQ   | 1.075       | 0.629       | 32.740       | 3.772        | 1.561       | 0.824       | 0.051        | 0.134        |
> | MMVP       | 1.655       | 0.789       | 36.791       | 4.243        | 2.517       | 1.187       | 0.051        | 0.155        |
> | MMVP+SVQ   | 1.563       | 0.751       | 35.837       | 4.133        | 2.470       | 1.107       | 0.051        | 0.155        |
>
> **Table: Additional baselines on TaxiBJ dataset**
>
> | Model          | MSE↓   | MAE↓   | SSIM↑   | PSNR↑   |
> |----------------|---------|---------|----------|----------|
> | WaST           | 0.3331  | 15.30   | 0.9835   | 39.60    |
> | WaST+SVQ       | 0.3329  | 14.96   | 0.9838   | 39.76    |
> | SwinLSTM       | 0.4256  | 16.95  | 0.9785   | 38.65    |
> | SwinLSTM+SVQ   | 0.3583  | 15.98   | 0.9815   | 39.14    |
>
> [1] Tan C, Li S, Gao Z, et al. Openstl: A comprehensive benchmark of spatio-temporal predictive learning[J]. Advances in Neural Information Processing Systems, 2023, 36: 69819-69831.
>
> [2] Zhong, Yiqi, et al. "Mmvp: Motion-matrix-based video prediction." Proceedings of the IEEE/CVF International Conference on Computer Vision. 2023.
>
> [3] Tang, Song, et al. "Swinlstm: Improving spatiotemporal prediction accuracy using swin transformer and lstm." Proceedings of the IEEE/CVF International Conference on Computer Vision. 2023.
>
> [4] Nie, X., et al. "Wavelet-Driven Spatiotemporal Predictive Learning: Bridging Frequency and Time Variations." Proceedings of the AAAI Conference on Artificial Intelligence, 2024, 38(5): 4334-4342.
>
> > **[Q3]** The explanation of the quantization module's implementation lacks depth regarding its integration with the overall spatio-temporal forecasting model. While the authors outline the architecture and components involved, they do not provide sufficient details on how the quantization process interacts with other model elements or influences the final forecasting results. A more thorough exploration of these interactions would enhance the clarity and applicability of their proposed method.
>
> We concur with the review's observation regarding the integration of the baseline method and the VQ method. Our goal is for our method to enhance various baseline methods, and digging into each baseline's interaction may not seem universally applicable. However, we fully understand your inquiry, as we intended to investigate this aspect during the project. Consequently, we explore the positioning issue of VQ in Section 5.4. Our findings indicate that performing quantization before the translator provides a significant optimization advantage, resulting in a smoother learning curve and more accurate outcomes compared to quantization after the translator. This suggests that our SVQ design should be closer to the original input prior to the baseline algorithm's representation learning module, indicating that a weak and pre-processed VQ block is sufficient to enhance performance without strong interaction with the base model.

---

> ### Author Response · Authors · 2024-11-29
> **Looking forward to your feedback**
>
> Dear reviewer AkZ2,
>
> We sincerely appreciate your time and effort in reviewing our work and offering valuable feedback. As the discussion period is nearing its end, we would like to confirm whether our responses have effectively addressed your concerns.
>
> If you require further clarification or have any additional concerns, please do not hesitate to contact us! We are more than willing to continue our communication with you. More discussions and suggestions on further improving the paper are also always welcomed!
>
> Best regards,
>
> The Authors

---

> > ### Author Response · Authors · 2024-12-03
> > **Copy the reviewer AkZ2's replying here**
> >
> > We are copying reviewer AkZ2's response here, as he replied at the top, to make the individual discussion easier to follow.
> >
> > AKZ2:  I have read the authors' responses, which helps me better understand their method. I tend to keep my rating.
> >
> > Ours:  Thank you for taking the time to review and respond to our work. It's fine to keep the score. we're glad that our response contributes to a better understanding.

---

### Official Review · Reviewer_pbDr · 2024-11-04

**Soundness:** 3
**Presentation:** 3
**Contribution:** 3
**Rating:** 8
**Confidence:** 3

**Summary:**

Vector quantization (VQ) is insufficient in improving the accuracy of spatiotemporal prediction. This paper introduces differentiable sparse soft vector quantization (SVQ) that can strike a balance between detail preservation and noise reduction, providing a solid foundation for full differentiability and sparse regression. Empirical studies on five spatiotemporal benchmark datasets show that SVQ achieves the best results, including a 7.9% improvement on the WeatherBench-S temperature dataset, a 9.4% average MAE reduction in video prediction benchmarks (Human3.6M, KTH, and KittiCaltech), and a 17.3% improvement in image quality (LPIPS).

**Strengths:**

1. The manuscript proposes a differentiable sparse soft vector quantization (SVQ) method, which is the first vector quantization method applied to spatiotemporal prediction and shows significant improvement.
2. The SVQ method proposed in the manuscript has achieved leading performance in multiple real-world spatiotemporal prediction tasks, significantly reducing errors on multiple benchmark datasets, such as reducing errors by 7.9% on the WeatherBench dataset.
3. The SVQ proposed in the manuscript can be seamlessly integrated into different types of spatiotemporal prediction models as a plug-in, and has improved performance in various architectures, demonstrating the versatility of the method.

**Weaknesses:**

1.The SVQ method proposed in the manuscript still requires a lot of computing resources, especially in the case of high-dimensional data and large-scale codebooks.
2.The comparison methods cited by the author in Tables 1 and 2 are only up to date in 2022, and lack comparisons of the latest methods in the past two years.

**Questions:**

1.Will the SVQ module added as a plug-in to the model have similar performance improvements for other tasks (such as image generation or natural language processing)? What are the applicable application scenarios?

---

> ### Author Response · Authors · 2024-11-23
> **Response to Reviewer pbDr**
>
> Dear Reviewer pbDr,
>
> We would like to express our gratitude to you for championing our work and for the thoughtful review, particularly in highlighting the weaknesses and questions section. We are committed to addressing these points and alleviating any concerns.
>
> > **[Q1]** The SVQ method proposed in the manuscript still requires a lot of computing resources, especially in the case of high-dimensional data and large-scale codebooks.
>
> Thank you for taking the time to examine the appendix section and verify the computational resource results. It might be beneficial to include these findings in the main section, as they were overlooked by another reviewer. You are correct in noting that SVQ increases the FLOPs. The introduction of our SVQ design results in a slight increase—less than 10\%—in the number of learnable parameters and a moderate increase in FLOPs, ranging from 30\% to 70\%, depending on the task.
>
> We would like to emphasize that the significant improvements observed here are largely attributable to the high efficiency of our chosen baseline model, SimVP. In our proposed method, the increase in learnable parameters and FLOPs is primarily determined by the codebook size and dimension, and is independent of the base model. If a more complex baseline with a larger parameter size and more FLOPs, such as ConvLSTM, PredRNN, or other recurrent-based models, were used, the increase in FLOPs would be much less than 2\%. Below we show the parameter size and FLOPs on a more complex baseline: MMVP [1]. Besides, our ablation experiments have shown that the codebook size can remain relatively small while still yielding significant improvements, so a smaller codebook size can be chosen if that's an issue.
>
> **Table: Computational cost of SVQ**
>
> | Model          | Params   | FLOPs   |
> |----------------|---------|---------|
> | MMVP           | 16.378M  | 33.637G   |
> | MMVP+SVQ       | 17.992M  | 34.253G   |
>
> [1] Zhong, Yiqi, et al. "Mmvp: Motion-matrix-based video prediction." Proceedings of the IEEE/CVF International Conference on Computer Vision. 2023.
>
> > **[Q2]** The comparison methods cited by the author in Tables 1 and 2 are only up to date in 2022, and lack comparisons of the latest methods in the past two years.
>
> Thank you for your suggestion. We are currently adhering to the OpenSTL benchmark [1], the ongoing benchmark for spatio-temporal forecasting. In light of this, we would like to include three additional baselines from the past two years: MMVP (ICCV 2023) [2], SwinLSTM (ICCV 2023) [3], and WaST (AAAI 2024) [4]. MMVP and WaST are evaluated on WeatherBench-S dataset. Due to SwinLSTM only supports H = W for image input, we evaluate it on TaxiBJ dataset. The results are presented in the table below. SVQ consistently enhances the performance of these baselines. Furthermore, given the general nature of our proposed method, which addresses non-differentiability and representation issues typically orthogonal to base model design, we believe it is likely to enhance future methodologies.
>
> **Table: Additional baselines on WeatherBench-S dataset**
>
> | Model      | Temp (MSE↓) | Temp (MAE↓) | Humid (MSE↓) | Humid (MAE↓) | Wind (MSE↓) | Wind (MAE↓) | Cloud (MSE↓) | Cloud (MAE↓) |
> |------------|-------------|-------------|--------------|--------------|-------------|-------------|--------------|--------------|
> | WaST       | 1.239       | 0.686       | 33.349       | 3.881        | 1.603       | 0.838       | 0.048        | 0.150  |
> | WaST+SVQ   | 1.075       | 0.629       | 32.740       | 3.772        | 1.561       | 0.824       | 0.051        | 0.134 |
> | MMVP       | 1.655       | 0.789       | 36.791       | 4.243        | 2.517       | 1.187       | 0.051        | 0.155   |
> | MMVP+SVQ   | 1.563       | 0.751       | 35.837       | 4.133        | 2.470       | 1.107       | 0.051        | 0.155 |
>
> **Table: Additional baselines on TaxiBJ dataset**
>
> | Model          | MSE↓   | MAE↓   | SSIM↑   | PSNR↑   |
> |----------------|---------|---------|----------|----------|
> | WaST           | 0.3331  | 15.30   | 0.9835   | 39.60    |
> | WaST+SVQ       | 0.3329  | 14.96   | 0.9838   | 39.76    |
> | SwinLSTM       | 0.4256  | 16.95  | 0.9785   | 38.65    |
> | SwinLSTM+SVQ   | 0.3583  | 15.98   | 0.9815   | 39.14    |
>
> [1] Tan C, Li S, Gao Z, et al. Openstl: A comprehensive benchmark of spatio-temporal predictive learning[J]. Advances in Neural Information Processing Systems, 2023, 36: 69819-69831.
>
> [2] Zhong, Yiqi, et al. "Mmvp: Motion-matrix-based video prediction." Proceedings of the IEEE/CVF International Conference on Computer Vision. 2023.
>
> [3] Tang, Song, et al. "Swinlstm: Improving spatiotemporal prediction accuracy using swin transformer and lstm." Proceedings of the IEEE/CVF International Conference on Computer Vision. 2023.
>
> [4] Nie, X., et al. "Wavelet-Driven Spatiotemporal Predictive Learning: Bridging Frequency and Time Variations." Proceedings of the AAAI Conference on Artificial Intelligence, 2024, 38(5): 4334-4342.

---

> ### Author Response · Authors · 2024-11-23
> **Continue Response to Reviewer pbDr**
>
> > **[Q3]** Will the SVQ module added as a plug-in to the model have similar performance improvements for other tasks (such as image generation or natural language processing)? What are the applicable application scenarios?
>
> We anticipate that our proposed method will likely improve image generation due to its broad applicability in addressing non-differentiability and representation issues that are generally unrelated to the design of base models. While image generation seems promising, the impact on NLP might differ, as most LLMs utilize quantization based on the natural segmentation of words without VQ blocks. We are eager to test our method in domains such as image generation, multimodal LLMs, and other application scenarios where existing VQ blocks provide a natural substitution. However, many of these tasks are extremely GPU-intensive, and we currently lack the resources to pursue them. Spatio-temporal forecasting, on the other hand, is an application that does not require tens or hundreds of GPUs, making it a viable option for exploration.

---

> > ### Comment · Reviewer_pbDr · 2024-11-26
> > **The feedback resolved my concerns**
> >
> > The feedback was excellent, it resolved all my concerns. I have updated my score to 8.

---

> > > ### Author Response · Authors · 2024-11-27
> > > **Thank you for your valuable feedback**
> > >
> > > Dear Reviewer pbDr,
> > >
> > > We are pleased to hear that our new experimental results and clarifications have positively contributed to the paper.
> > >
> > > Thank you for your insightful review, which has significantly improved the quality of our work.
> > >
> > > Kind regards,
> > >
> > > The Authors

---

### Official Review · Reviewer_PA7x · 2024-11-04

**Soundness:** 2
**Presentation:** 1
**Contribution:** 2
**Rating:** 5
**Confidence:** 4

**Summary:**

This paper identifies the limited performance of traditional vector quantization (VQ) in spatiotemporal forecasting due to non-differentiability and limited representation power. It proposes Differentiable Sparse Soft - Vector Quantization (SVQ), which approximates sparse regression with a two-layer MLP for differentiability and uses Soft-VQ with sparse regression to capture patterns and filter noise. Empirical results on multiple datasets show SVQ achieves state-of-the-art performance, is versatile as a plug-in, and effectively balances detail and noise reduction. Ablation studies and additional analyses confirm its key components’ importance and its robustness.

**Strengths:**

The paper shows originality through the development of SVQ, a novel combination of sparse regression and soft vector quantization for spatio-temporal forecasting with theoretical analysis. Empirical validation is extensive, with experiments on multiple real - world datasets and comparisons to existing methods, achieving state-of-the-art results and validating the method's effectiveness and quality. It has potential applications in various domains and can inspire future research, opening new avenues for exploration and providing insights for model development.

**Weaknesses:**

1. The novelty of the approach is limited, and the performance improvement appears marginal. Furthermore, there is no discussion on the additional computational overhead that these slight performance enhancements (such as those observed in wind speed and humidity in Table 1, and in the KTH dataset in Table 2, as well as in Table 18) will incur. A more detailed analysis of the computational overhead introduced by SVQ, compared to baseline methods, is needed, especially for cases where the performance gains are smaller.

2. The motivation is unclear. It is not surprising to use huge codebook and sparse representation to improve the effect, because huge codebook itself brings a lot of extra parameter redundancy. So why use vector quantization? Because in other fields (e.g., video compression, video generation), VQ is to compress redundant information, not to add redundant information. You could further explain their rationale for using vector quantization in this context, given its typical use for compression in other fields.

3. The theoretical analysis provided seems unrelated to the content of the article. Furthermore, the article fails to discuss the relationship between the information or features extracted after compression using SVQ and the original spatiotemporal data. Consequently, there is a notable lack of corresponding theoretical discussion. A more in-depth exploration of how the features learned through SVQ are related to the original spatiotemporal data would greatly aid in fully elucidating the mechanism underlying SVQ.

4. The ablation study conducted is insufficient. There is no doubt that setting the code size to 10000 will yield better performance compared to 1000. A more detailed discussion of the trade-offs involved (such as efficiency, convergence, etc.) with larger code sizes would be helpful.

5. The introduction of redundant over-complete codebooks and additional computational overhead has resulted in a lack of discussion on computational efficiency, speed, and complexity, among other factors. Empirical measurements of training and inference times, memory usage, and computational complexity, as a function of codebook size, would provide a more comprehensive illustration of the advantages of SVQ.

**Questions:**

1. What is the motivation of using vector quantization into spatiotemporal prediction?
2. What is the significance of theoretical analysis in Chapter 4? Is this theoretical analysis related to video prediction?
3. What are the ''improvement'' in Table 1.2.3. refer to, SimVP?
4. Can the method proposed in the paper be compared with diffusion based models? For example, ExtDM: Distribution Extrapolarization Diffusion Model for Video Prediction, CVPR2024. What are the differences between these two methods, e.g., their application scenarios or efficiency?
5. Why are there different types of comparison results between WeatherBench-S and WeatherBench-M in Tab. 1 (Total Cloud Cover in WeatherBench-S and Wind UV in WeatherBench-M)? Why not compare the same subjects? What are the differences in SVQ performance across different physical quantities and data scales?

---

> ### Author Response · Authors · 2024-11-23
> **Response to Reviewer PA7x**
>
> Dear Reviewer PA7X,
>
> Thank you for your thorough analysis and constructive feedback on our paper. Before addressing the specifics of your questions, we would like to provide some context regarding the weaknesses you highlighted. We are grateful to you for raising these important points, as they were considerations we also had before commencing our project. Allow us to explain them in detail below.
>
> > **[Q1]** Regarding the motivation.
>
> As you insightfully pointed out, it might seem intuitive that using an extensive codebook with numerous parameters would improve results. Initially, we shared this assumption. However, our findings reveal that this belief does not hold true for the majority of state-of-the-art VQ methods, as demonstrated in Table 4 and Figure 5 on page 8 of our paper (we put the table below to highlight as well). We conducted experiments by varying the size of the codebook, from small to large, and found that none led to improved outcomes. Although a larger codebook size results in less deterioration, it does not enhance results. This has led us to conclude that there is an underlying issue that needs attention, and merely increasing the codebook size does not result in improvements compared to the non-VQ counterpart.
>
> **Table 4: Comparison of vector quantization methods**
>
> | Method                          | MSE ↓ | MAE ↓ |
> |---------------------------------|-------|-------|
> | Baseline (SimVP w/o VQ)         | 1.105 | 0.6567 |
> | VQ                              | 1.854 | 0.8963 |
> | Residual VQ                     | 1.213 | 0.6910 |
> | Grouped Residual VQ             | 1.174 | 0.6747 |
> | Multi-headed VQ                 | 1.211 | 0.6994 |
> | Stochastic Residual VQ          | 1.888 | 0.9237 |
> | Residual Finite Scalar Quantization | 1.319 | 0.7505 |
> | Lookup Free Quantization  | 2.988 | 1.1103 |
> | Residual LFQ             | 1.281 | 0.7281 |
> | SVQ-raw                         | 1.123 | 0.6456 |
> | SVQ                             | 1.018 | 0.6109 |
>
> You are absolutely correct in noting that VQ compresses redundant information. In fact, we view VQ as a process that eliminates noise and retains the signal. We believe it is a general approach that can be applied to enhance prediction performance by improving the signal-to-noise ratio. Typically, the VQ method is effective in achieving this.
>
> **What we found** is that the non-differentiability of VQ introduces quantization error, or optimization error, during model training. This creates a trade-off between the deterioration caused by quantization error and the improvement achieved by VQ's ability to compress redundant information. To address this, we designed a differentiable VQ method that minimizes quantization error while focusing on enhancing prediction performance by increasing the signal-to-noise ratio.
>
> Another crucial observation is that the specific parameters, or learnable parameters, aren't as significant as one might expect. We conducted a frozen codebook experiment, which demonstrated that a randomly initialized frozen codebook almost achieved the same level of improvement as a learnable one as shown in Tables 1 and 2 (we put the table below). This finding supports the previous belief that parameters are not the key factor, and that merely increasing the size of the codebook doesn't necessarily enhance performance.
>
> **Table 1: WeatherBench-S results**
>
> | Model|Temp (MSE↓)|Temp (MAE↓)|Humid (MSE↓)|Humid (MAE↓)|Wind (MSE↓)|Wind (MAE↓)|Cloud (MSE↓)|Cloud (MAE↓)|
> |------|-----------|-----------|------------|------------|-----------|-----------|------------|------------|
> | SimVP (w/o VQ)|1.105|0.6567|31.332|3.776|1.4996|0.8145|0.0466|0.1469|
> | SimVP+SVQ (Frozen codebook)|1.023|0.6131|30.863|3.661|1.4337|0.7861|0.0456|0.1456|
> | SimVP+SVQ (Learnable codebook)|1.018|0.6109|30.611|3.657|1.4186|0.7858|0.0458|0.1463|
>
> **Table 2: Video prediction results**
>
> | Model       | Human3.6M      |        |      |        | KittiCaltech   |       |        |       | KTH           |      |       |        |
> |---------------|----------------|-------------|-------------|-------------|----------------|--------------|--------------|--------------|---------------|-------------|-------------|-------------|
> |               | MAE↓         | SSIM↑      | PSNR↑      | LPIPS↓     | MAE↓         | SSIM↑       | PSNR↑       | LPIPS↓      | MAE↓         | SSIM↑      | PSNR↑      | LPIPS↓     |
> | SimVP (w/o VQ)| 1441.0        | 0.9834      | 34.08       | 0.03224     | 1507.7        | 0.9453       | 27.89       | 0.05740     | 397.1         | 0.9065      | 27.46      | 0.26496     |
> | SimVP+SVQ (Frozen codebook)| 1264.9    | 0.9851      | 34.07       | 0.02380     | 1408.6        | 0.9469       | 28.10       | 0.05535     | 364.6         | 0.9109      | 27.28      | 0.20988     |
> | SimVP+SVQ (Learned codebook)| 1265.1   | 0.9851      | 34.06       | 0.02367     | 1414.9        | 0.9458       | 28.10       | 0.05776     | 360.2         | 0.9116      | 27.37      | 0.20658     |

---

> ### Author Response · Authors · 2024-11-23
> **Continue Response to Reviewer PA7x**
>
> > **[Q2]** Discussion on computational cost.
>
> Due to space constraints, the computational cost table has been relocated to Appendix Table 11. The introduction of our SVQ design results in a slight increase—less than 10\%—in the number of learnable parameters and a moderate FLOPs increase of 30\% to 70\%, depending on the task. While this might initially seem significant, our experimental results suggest otherwise. Our fully differentiable SVQ design, implemented with an MLP, extends the per-epoch training time on a V100 GPU by only about 15\% compared to the baseline with identical parameter settings but without VQ. Additionally, it offers a speed advantage over other VQ methods in most cases due to its differentiable nature. Consequently, it proves to be a highly efficient add-on block.
>
> We would like to emphasize that the significant improvements observed here are largely attributable to the high efficiency of our chosen baseline model, SimVP. In our proposed method, the increase in learnable parameters and FLOPs is primarily determined by the codebook size and dimension, and is independent of the base model. If a more complex baseline with a larger parameter size and more FLOPs, such as ConvLSTM or PredRNN, or other recurrent-based models, were used, the increase in FLOPs would be much less than 2\%. Below we show the parameter size and FLOPs on a more complex baseline: MMVP [1].
>
> **Table: Computational cost of SVQ**
>
> | Model          | Params   | FLOPs   |
> |----------------|---------|---------|
> | MMVP           | 16.378M  | 33.637G   |
> | MMVP+SVQ       | 17.992M  | 34.253G   |
>
> [1] Zhong, Yiqi, et al. "Mmvp: Motion-matrix-based video prediction." Proceedings of the IEEE/CVF International Conference on Computer Vision. 2023.
>
> > **[Q3]** Regarding novelty.
>
> Our innovation lies in addressing a challenge that no previous state-of-the-art VQ method has tackled—enhancing spatio-temporal forecasting results. We acknowledge that there are diverse opinions on the novelty of architectural improvements and whether they result in marginal benefits for a given task. However, we aim to address a binary question: does our approach improve or hinder performance? Furthermore, we investigate why such binary outcomes occur, particularly focusing on non-differentiable problems. This is how we define and position our contribution.
>
> > **[Q4]** Significance of theoretical analysis.
>
> Our theoretical analysis shows how the proposed VQ mimics the sparse regression model (i.e. ReLU and two layers in MLP correspond exactly to thresholding and quadratic dependence on fixed dictionary in sparse regression), and thus enjoys the advantage of sparse regression (i.e. error depends on $\log d$ not $d$), making it surpass traditional cluster-based VQ methods. Another key insight is that a frozen codebook can be nearly as effective as a trainable one, as long as it is sufficiently large. Although it may appear counterintuitive, our proof is based on a simple fact that any vector in a simplex can be well approximated by a sparse combination of vectors from a frozen and large dictionary. Thus, when using high-dimensional vectors, a sparse representation can be more expressive than a dense one in lower-dimensional space. Furthermore, for high-dimensional vectors, a basis can often be randomly sampled due to the high probability of orthogonality, eliminating the need for it to be learned. I hope this helps clarify the theoretical analysis, as these concepts are crucial for addressing the underlying 'why' questions.

---

> ### Author Response · Authors · 2024-11-23
> **Continue Response to Reviewer PA7x**
>
> **Question about some details**
>
> > **[Q5]** What does the "improvement" in Table 1.2.3 refer to, SimVP?
>
> Yes, that's compared to its SimVP backbone without adding SVQ.
>
> Perhaps there may be some misunderstanding regarding our method's performance. For example, as noted in Weakness 1: "slight enhancements ... in the KTH dataset in Table 2," our SVQ method actually leads to significant improvements in the KTH dataset results, with a 9.3% improvement in the MAE metric and a 22.0% improvement in the LPIPS metric. The related results are presented below.
>
> **Table 2: Results on KTH Dataset**
>
> | Model | MAE↓  | SSIM↑  | PSNR↑ | LPIPS↓ |
> |-------------------------------|-------|--------|-------|--------|
> | SimVP (w/o VQ) | 397.1 | 0.9065 | 27.46| 0.26496|
> | SimVP+SVQ (Frozen Codebook)| 364.6 |0.9109|27.28|0.20988|
> | SimVP+SVQ (Learned Codebook)| 360.2 |0.9116|27.37|0.20658|
> | Improvement| **9.3%↑** | **0.6%↑**  | 0.3%↓ | **22.0%↑**|
>
>
> > **[Q6]** Compared to diffusion methods.
>
> We concur with the knowledgeable reviewer that diffusion models are achieving state-of-the-art results in video forecasting tasks. Our work aligns with the NeurIPS 2023 benchmark paper, OpenSTL [1], and follows its ongoing updates to evaluate spatio-temporal forecasting baselines. This task encompasses three main categories: weather forecasting, video forecasting, and traffic forecasting. Video forecasting often emphasizes the assessment of output image quality, going beyond traditional metrics such as MSE and MAE, which have historically been the primary measures in both weather and traffic forecasting. Therefore, we opted to follow this setting.
>
> But we would like to include three additional baselines from the past two years which are included in the latest OpenSTL benchmark:  MMVP (ICCV 2023) [2], SwinLSTM (ICCV 2023) [3], and WaST (AAAI 2024) [4]. MMVP and WaST are evaluated on WeatherBench-S dataset. Due to SwinLSTM only supports H = W for image input, we evaluate it on TaxiBJ dataset. The results are presented in the table below. SVQ consistently enhances the performance of these baselines. Given the general nature of our proposed method, which addresses non-differentiability and representation issues that are largely independent of base model design, we believe it is likely to enhance future methodologies. Further research may be required to explore the application of our proposed method in diffusion models, given the complex interactions between noise and signal inherent in the diffusion process. Our findings in Appendix D.3 suggest that applying quantization before the translator module offers a significant optimization advantage. This approach results in a smoother learning curve and more accurate outcomes compared to quantization applied after the translator module. Consequently, our SVQ design should remain closely aligned with the original input, prior to the baseline algorithm's representation learning phase. This implies that a minimal and pre-processed VQ block is adequate for enhancing performance without extensive interaction with the base model. Therefore, VQ may not naturally align with the diffusion algorithm.
>
> **Table: Additional baselines on WeatherBench-S dataset**
>
> | Model | Temp (MSE↓) | Temp (MAE↓) | Humid (MSE↓) | Humid (MAE↓) | Wind (MSE↓) | Wind (MAE↓) | Cloud (MSE↓) | Cloud (MAE↓) |
> |------------|-------------|-------------|--------------|--------------|-------------|-------------|--------------|--------------|
> | WaST  | 1.239| 0.686  | 33.349 | 3.881 | 1.603| 0.838 | 0.048  | 0.150  |
> | WaST+SVQ   | 1.075 | 0.629| 32.740  | 3.772| 1.561  | 0.824  | 0.051  | 0.134  |
> | MMVP  | 1.655 | 0.789  | 36.791  | 4.243   | 2.517 | 1.187 | 0.051 | 0.155    |
> | MMVP+SVQ   | 1.563 | 0.751 | 35.837 | 4.133 | 2.470 | 1.107   | 0.051    | 0.155  |
>
>
> **Table: Additional baselines on TaxiBJ dataset**
>
> | Model          | MSE↓   | MAE↓   | SSIM↑   | PSNR↑   |
> |----------------|---------|---------|----------|----------|
> | WaST     | 0.3331  | 15.30   | 0.9835| 39.60|
> | WaST+SVQ  | 0.3329  | 14.96 | 0.9838 | 39.76|
> | SwinLSTM | 0.4256  | 16.95  | 0.9785  | 38.65|
> | SwinLSTM+SVQ| 0.3583  | 15.98   | 0.9815   | 39.14 |
>
> [1] Tan C, Li S, Gao Z, et al. Openstl: A comprehensive benchmark of spatio-temporal predictive learning[J]. Advances in Neural Information Processing Systems, 2023, 36: 69819-69831.
>
> [2] Zhong, Yiqi, et al. "Mmvp: Motion-matrix-based video prediction." Proceedings of the IEEE/CVF International Conference on Computer Vision. 2023.
>
> [3] Tang, Song, et al. "Swinlstm: Improving spatiotemporal prediction accuracy using swin transformer and lstm." Proceedings of the IEEE/CVF International Conference on Computer Vision. 2023.
>
> [4] Nie, X., et al. "Wavelet-Driven Spatiotemporal Predictive Learning: Bridging Frequency and Time Variations." Proceedings of the AAAI Conference on Artificial Intelligence, 2024, 38(5): 4334-4342.

---

> ### Author Response · Authors · 2024-11-23
> **Continue Response to Reviewer PA7x**
>
> > **[Q7]** Why are there different types of comparison results between WeatherBench-S and WeatherBench-M in Tab. 1 (Total Cloud Cover in WeatherBench-S and Wind UV in WeatherBench-M)? Why not compare the same subjects?
>
> Our work aligns with the NeurIPS 2023 benchmark paper, OpenSTL [1]. OpenSTL provides both single-channel and multi-channel forecasting options for WeatherBench. WeatherBench-S involves predicting at 1-hour intervals, while WeatherBench-M operates at 6-hour intervals and larger dataset. This setup is defined within OpenSTL to evaluate models across a broader range of temporal and spatial complexities. Therefore, comparing the same subjects across these two datasets would deviate from the benchmark.
>
> [1]Tan C, Li S, Gao Z, et al. Openstl: A comprehensive benchmark of spatio-temporal predictive learning[J]. Advances in Neural Information Processing Systems, 2023, 36: 69819-69831.
>
> > **[Q8]** How are the differences in SVQ performance across different physical quantities and data scales?
>
> We conducted a high-dimensional experiment detailed in Appendix D.8, Table 19, using 110 weather covariates to demonstrate the effect on various physical quantities. Of these, 109 showed positive improvement with the addition of SVQ, with an average improvement of over 8.9\%. These findings underscore that despite the complexities introduced by high-dimensional datasets, SVQ effectively adapts to and enhances the predictive capabilities of the backbone model. Notably, we observed significant improvements in channels such as temperature at different heights, geopotential height, and potential vorticity. The improvements were particularly pronounced for high altitude features associated with low pressure height, while surface weather features exhibited comparatively lower improvements.
>
> Since all variables are normalized before being input into the prediction algorithm, we do not observe a clear difference between channels with varying scales. However, we are not entirely sure if the question pertains to channel scale or dataset size. Therefore, we conducted additional few-sample experiments using the Weatherbench-S-temperature dataset, with SimVP as our baseline model:
>
> |With SVQ|MSE|MAE|RMSE|
> |------|------|------|------|
> |10\%|1.129|0.661|1.063|
> |30\%|1.036|0.615|1.018|
> |50\%|1.036|0.612|1.018|
> |70\%|1.030|0.611|1.015|
> |90\%|1.022|0.611|1.011|
>
> |Without SVQ|MSE|MAE|RMSE|
> |------|------|------|------|
> |10\%|1.192|0.684|1.092|
> |30\%|1.166|0.674|1.080|
> |50\%|1.141|0.661|1.068|
> |70\%|1.223|0.700|1.106|
> |90\%|1.118|0.659|1.057|
>
>
> The results show that SVQ consistently enhances its performance across different percentages in few-sample experiments. When 10\% of the data is used, the improvement is about 5\%. This enhancement stabilizes at approximately 12\% as the data percentage increases from 30\% to 90\%.

---

> ### Author Response · Authors · 2024-11-29
> **Looking forward to your feedback**
>
> Dear reviewer PA7x,
>
> Thank you very much for taking the time to review our work and for providing valuable feedback. We were wondering if our responses have resolved your concerns.
>
> We will be happy to have further discussions if there are still some remaining questions! More discussions and suggestions on further improving the paper are also always welcomed!
>
> We look forward to hearing from you and remain available to provide any additional details that might assist in resolving your concerns.
>
> Best regards,
>
> The Authors

---

> > ### Comment · Reviewer_PA7x · 2024-12-02
> > **Response to Authors**
> >
> > Thank you for your detailed reply and comment, which addressed part of my concern (the additional calculation redundancy). I will maintain my score.
> > SVQ appears to be a promising VQ technology that resolves some of the issues in existing VQ technologies, yet I still fail to see the necessity of SVQ in video prediction tasks. In other words, the common and unremarkable approach of enhancing performance through increased parameters and computation applies here as well. However, what specific challenges in video prediction necessitate the use of VQ or its differentiable variant, SVQ? Furthermore, when compared to eliminating noise, restoring high-frequency details poses a greater difficulty in video prediction, generation, reconstruction, and compression tasks (spectral bias). Regarding the effectiveness of SVQ, particularly addressing the issues of "inaccurate optimization due to non-differentiability and limited representation power in hard VQ", it would be more appropriate to discuss these in the contexts of image/video restoration, generation, and compression, rather than solely focusing on video prediction.

---

> ### Author Response · Authors · 2024-12-02
> **Response to Reviewer PA7x**
>
> Dear reviewer PA7x,
>
> Thank you so much for your response. We completely understand and appreciate your decision to maintain the score, and we are grateful for the time you spent reviewing our work and helping us improve it. We're pleased to hear that our additional experiment addressed some of your concerns.
>
> In our experiments, video prediction is indeed not the only task. As our work follows OpenSTL benchmark, which includes three tasks: weather forecasting, traffic forecasting, and video prediction. We agree with you that video prediction, compared to the other two tasks, tends to focus more on the quality of video generation, specifically how realistic the generated images are. On the other hand, the weather and traffic forecasting tasks, which use MSE/MAE metrics, may align more closely with our design approach—denoising and preserving the signal. On this point, we are aligned with your perspective.

---

> ### Author Response · Authors · 2024-12-02
> **Further Clarifications on Reviewer PA7x's Questions**
>
> Dear reviewer PA7x,
>
> We sincerely appreciate the time and effort you have dedicated to reviewing our work and for raising insightful concerns. We would like to take this opportunity to further clarify some possible misunderstanding.
>
> > **[1]** "In other words, the common and unremarkable approach of enhancing performance through increased parameters and computation applies here as well. "
>
> It might seem intuitive that using a large codebook with numerous parameters would naturally lead to improved results. Initially, we shared this assumption. However, our findings demonstrate that this belief does not hold true for the majority of state-of-the-art VQ methods, as evidenced by Table 4 and Figure 5 on page 8 of our paper (as previously noted in our first rebuttal). We conducted experiments by varying the size of the codebook, from small to large, and found that none led to improved outcomes.
>
> Therefore, the enhancements observed in our method cannot be attributed significantly, if at all, to increased parameters or computational overhead. Instead, the improvements are primarily attributed to two key factors: first, the differentiable SVQ design, which minimizes quantization error, and second, the enhanced representation power achieved through sparse regression-based VQ.
>
> > **[2]** "However, what specific challenges in video prediction necessitate the use of VQ or its differentiable variant, SVQ?"
>
> Our primary motivation lies in leveraging VQ as a process to reduce noise while preserving the signal. We view this as a general approach applicable to a variety of spatio-temporal forecasting tasks to enhance prediction performance by improving the signal-to-noise ratio. While video prediction task may often emphasize the quality of video generation, SVQ is designed with broader applicability in mind. It aims to address challenges across diverse spatio-temporal forecasting tasks. These tasks frequently require a balance between denoising and preserving meaningful details, a need that aligns closely with the core motivation behind SVQ. Our experimental results in weather forecasting and traffic forecasting further validate this point.
>
> If you need further clarification or have any additional questions, please don’t hesitate to contact us. We are more than willing to continue the discussion.
>
> Best regards,
>
> The Authors

---

### Comment · Reviewer_AkZ2 · 2024-12-03
**read authors' responses**

I have read the authors' responses, which helps me better understand their method. I tend to keep my rating.

---

> ### Author Response · Authors · 2024-12-03
>
> Thank you for taking the time to review and respond to our work. It's fine to keep the score. we're glad that our response contributes to a better understanding.

---

### Author Response · Authors · 2024-12-03
**Author's Summary on the Rebuttal**

We sincerely thank all reviewers for their valuable feedback that helped improve our paper. We also sincerely thank the AC in advance for the considerable time and effort devoted to further reviewing and facilitating the discussion process.

---
> **Summary of Discussion Process**

+ **Resolved Concerns**:

  + Reviewer **PA7x**: The main concerns were the motivation, theoretical analysis, and computational cost. Although the motivation was not fully confirmed, we believe that the latter two concerns were addressed after discussion.

  + Reviewer **pbDr**: The main concerns were the computational cost and the lack of comparisons with the latest methods from the past two years. The reviewer confirmed that all concerns were effectively resolved after discussion.

  + Reviewer **AkZ2**: The main concerns were the theoretical advantages of SVQ and the lack of comparisons with the latest methods from the past two years. These concerns were effectively resolved after discussion. The reviewer appreciated our responses, which helped them better understand our method.

  + Reviewer **1Zsw**: The main concern was the visual layout of the paper. We provided a revised PDF with an optimized layout, and we believe that the layout issue has been addressed.

+ **Remaining Concerns**:

  + Reviewer **PA7x**: The reviewer appreciated that SVQ appears to be a promising VQ technology that addresses some of the issues in existing VQ technologies. But the remaining concerns are:
      + The unclear motivation for applying VQ to video prediction tasks and why we only experimented with video prediction.

      We have clarified that video prediction is not the only task, as we also experimented with other two tasks—weather forecasting and traffic forecasting. Our primary motivation lies in leveraging VQ as a process to reduce noise while preserving the signal. We view this as a general approach applicable to a variety of spatio-temporal forecasting tasks to enhance prediction performance by improving the signal-to-noise ratio. While video prediction task may often emphasize the quality of video generation, SVQ is designed with broader applicability in mind. It aims to address challenges across diverse spatio-temporal forecasting tasks. These tasks frequently require a balance between denoising and preserving meaningful details, a need that aligns closely with the core motivation behind SVQ. Our experimental results in weather forecasting and traffic forecasting further validate this point.

      + The use of a common and unremarkable approach to enhance performance through increased parameters and computation.

      We clarified that this may be a misunderstanding, as the enhancements observed in our method cannot be attributed significantly, if at all, to increased parameters or computational overhead. Instead, the improvements are primarily attributed to two key factors: first, the differentiable SVQ design, which minimizes quantization error, and second, the enhanced representation power achieved through sparse regression-based VQ.

---

### Meta-Review · Area_Chair_hDNG · 2024-12-19

**Metareview:**

The paper was previously submitted to NeruIPS 2024, with title “A Differentiable Sparse Soft-Vector Quantization (SVQ) for Spatio-Temporal Forecasting”, but got rejected due to some weaknesses. The authors didn’t make material improvements except chancing the title into “Does Vector Quantization Fail in Spatio-Temporal Forecasting? Exploring a Differentiable Sparse Soft-Vector Quantization Approach”. Hence, I would recommend rejecting the paper.

Some comments for the authors to consider:

1.  The proposed Sparse Soft-Vector Quantization (SVQ) is not specific to spatio-temporal forecasting, but instead a general compression method that can be used for various tasks. The authors consider only spatio-temporal forecasting for experimental evaluation though. Since spatio-temporal forecasting is a comprehensive task that can be affected by many components, the presented results are not convincing to demonstrated the superiority of SVQ over its competitors. On the other hand, while evaluating the paper from the viewpoint of spatio-temporal forecasting, the link between SVQ and the problem space is unclear, as pointed out by Reviewer PA7x and Reviewer AkZ2.

2. It seems that the theorems presented in Section 4 are specific to the traditional SVQ, not the proposed differentiable SVQ.

3. What do you mean by $R_+^m$? Is it to enforce all components being nonnegative? If so, the deductions in Eq(2) and Eq(3) are wrong.

**Additional Comments On Reviewer Discussion:**

After author-reviewer discussions, Reviewer PA7x indicates:
“SVQ appears to be a promising VQ technology that resolves some of the issues in existing VQ technologies, yet I still fail to see the necessity of SVQ in video prediction tasks. In other words, the common and unremarkable approach of enhancing performance through increased parameters and computation applies here as well. However, what specific challenges in video prediction necessitate the use of VQ or its differentiable variant, SVQ? Furthermore, when compared to eliminating noise, restoring high-frequency details poses a greater difficulty in video prediction, generation, reconstruction, and compression tasks (spectral bias). Regarding the effectiveness of SVQ, particularly addressing the issues of "inaccurate optimization due to non-differentiability and limited representation power in hard VQ", it would be more appropriate to discuss these in the contexts of image/video restoration, generation, and compression, rather than solely focusing on video prediction.”


AC:  I would suggest the authors considering my comments into revision. Otherwise, such concerns may continue in future submission.

---

### Decision · Program_Chairs · 2025-01-22

Reject